

# The global aerosol-climate model ECHAM6.3-HAM2.3 – Part 2: Cloud evaluation, aerosol radiative forcing and climate sensitivity

David Neubauer[1], Sylvaine Ferrachat[1], Colombe Siegenthaler-Le Drian[1], Philip Stier[2], Daniel G. Partridge[3], Ina Tegen[4], Isabelle Bey[1,a], Tanja Stanelle[1], Harri Kokkola[5] and Ulrike Lohmann[1]

[1]Institute of Atmospheric and Climate Science, ETH Zurich, Zurich, Switzerland
[2]Department of Physics, University of Oxford, Oxford, United Kingdom
[3]College of Engineering, Mathematics, and Physical Sciences, University of Exeter, Exeter, United Kingdom
[4]Leibniz Institute for Tropospheric Research, Leipzig, Germany
[5]Finnish Meteorological Institute, Kuopio, Finland
[a]now at: MeteoSwiss, Geneva, Switzerland

*Correspondence to:* David Neubauer (david.neubauer@env.ethz.ch)

**Abstract.** The global aerosol-climate model ECHAM6.3-HAM2.3 (E63H23) and the previous model versions ECHAM5.5-HAM2.0 (E55H20) and ECHAM6.1-HAM2.2 (E61H22) are evaluated using global observational datasets for clouds and precipitation. In E63H23 low cloud amount, liquid and ice water path and cloud radiative effects are more realistic than in previous model versions. E63H23 has a more physically based aerosol activation scheme, improvements in the cloud cover scheme, changes in detrainment of convective clouds, changes in the sticking efficiency for accretion of ice crystals by snow, consistent ice crystal shapes throughout the model, changes in mixed phase freezing and an inconsistency in ice crystal number concentration (ICNC) in cirrus clouds was removed. Biases that were identified in E63H23 (and in previous model versions) are a too low cloud amount in stratocumulus regions, deep convective clouds in the Atlantic and Pacific oceans form too close to the continents and there are indications that ICNCs are overestimated.

Since clouds are important for effective radiative forcing due to aerosol-radiation and aerosol-cloud interactions (ERF$_{ari+aci}$) and equilibrium climate sensitivity (ECS), also differences in ERF$_{ari+aci}$ and ECS between the model versions were analyzed. ERF$_{ari+aci}$ is weaker in E63H23 (-1.0 W m$^{-2}$) than in E61H22 (-1.2 W m$^{-2}$) (or E55H20; -1.1 W m$^{-2}$). This is caused by the weaker shortwave ERF$_{ari+aci}$ (new aerosol activation scheme and sea salt emission parameterization in E63H23, more realistic simulation of cloud water) overcompensating the weaker longwave ERF$_{ari+aci}$ (removal of an inconsistency in ICNC in cirrus clouds in E61H22).

The decrease in ECS in E63H23 (2.5 K) compared to E61H22 (2.8 K) is due to changes in the entrainment rate for shallow convection (affecting the cloud amount feedback) and a stronger cloud phase feedback.

## 1 Introduction

Clouds are the largest modulators of radiation in Earth's atmosphere. Cloud hydrometeors are generally shorter-lived than other modulators of radiation in the atmosphere like aerosol particles, greenhouse gases or changes in surface albedo through



changes in land use. Also, the spatial structure of multiple clouds shows a large variability on different scales as it depends not only on large scale motions of the air but also on convective and turbulent motions at different scales. The range of microphysical properties of cloud droplets and ice crystals adds to the complexity of clouds in Earth's atmosphere. This complexity makes clouds difficult to observe and to simulate using models, resulting in the current large uncertainties in

future climate projections. Therefore, it is necessary to have a realistic representation of clouds in global climate models to be able to study past and present climate forcings and to strengthen the reliability of climate projections. It is crucial to evaluate clouds in these models with reliable observations and account for the complexity in clouds in the process.

In this study we use current satellite products to evaluate the aerosol-climate model ECHAM6.3-HAM2.3 and the two precursor model versions ECHAM6.1-HAM2.2 and ECHAM5.5-HAM2.0. One problem in using satellite products is that

they are produced with retrieval algorithms that have to make assumptions about the nature of the clouds (e.g. assumptions about the vertical cloud profile; Miller et al., 2016) (and other modulators of radiation) which will not always fit optimally for every cloud in the observed satellite pixels. Accordingly, current satellite products include measures of uncertainty in the retrieved cloud properties. We use these uncertainty measures to limit the evaluation only to regions where the observations are reliable. Furthermore, we apply the CFMIP (Cloud Feedback Model Intercomparison Project) Observation Simulator

Package (COSP) where appropriate, to account for limitations in the satellite observations (e.g. clouds cannot be observed below the level of full lidar signal attenuation by spaceborne lidar; Chepfer et al., 2010), the different scales of the model grid compared to the satellite data and ensure to compare exactly the same variables in the model output as in the satellite products.

To further limit the impact of observational uncertainties we use several products from independent instruments and aim at

identifying model biases in several of them. We also perform some of the analysis for different regions to study biases for different cloud types.

For studying past and present climate forcings it is indispensable to constrain the effective anthropogenic aerosol forcing due to aerosol-radiation and aerosol-cloud interactions ($ERF_{ari+aci}$). Because of the large impact of clouds on radiation, the representation of clouds in a global model can have an impact on $ERF_{ari+aci}$. Therefore, we also investigate the difference in

$ERF_{ari+aci}$ in the three ECHAM-HAM model versions and how they compare to differences in the simulations of clouds. As cloud feedbacks will have a large impact on temperature in a warmer climate we compare equilibrium climate sensitivity (ECS) and cloud feedbacks of the different model versions.

Section 2 gives a short description of the representation of clouds in ECHAM6.3-HAM2.3 and of the observational products applied in the model evaluation. In section 3 the results of the cloud evaluation, the comparison of $ERF_{ari+aci}$, $RF_{ari}$ and ECS

in the ECHAM-HAM model versions are presented and discussed. The results are summarized in section 4 and conclusions are drawn.



## 2 Methodology

### 2.1 Model description

The global aerosol-climate model ECHAM-HAM is the combination of the global climate model ECHAM with the aerosol microphysics module HAM (Stier et al., 2005). The ECHAM5 and ECHAM6 model versions used in this study are described in Roeckner et al. (2003) and Stevens et al. (2013) respectively. The ECHAM-HAM model versions and configurations used in this study are described in separate studies: ECHAM5.5-HAM2.0 in Zhang et al. (2012), ECHAM6.1-HAM2.2 in Neubauer et al. (2014) and ECHAM6.3-HAM2.3 in Tegen et al. (2018). For the sake of brevity, in the following ECHAM5.5-HAM2.0 will be referred to as E55H20, ECHAM6.1-HAM2.2 as E61H22 and ECHAM6.3-HAM2.3 as E63H23. In contrast to the 1-moment cloud microphysics scheme in the ECHAM base model (Lohmann and Roeckner, 1996), ECHAM-HAM uses a 2-moment cloud microphysics scheme. The 2-moment cloud microphysics scheme is described in Lohmann et al. (2007), Lohmann and Hoose (2009) and recent changes and improvements applied in E63H23 in Lohmann and Neubauer (2018). A 2-moment cloud microphysics scheme is required to study aerosol-cloud interactions. In ECHAM-HAM cloud droplet activation and ice crystal nucleation from cloud condensation nuclei and ice nucleating particles are computed as well as in-cloud and below cloud scavenging. Therefore, ECHAM-HAM simulates aerosol-cloud interactions in liquid, mixed-phase and ice clouds. However, a 2-moment cloud microphysics scheme is not only a prerequisite for simulating aerosol-cloud interactions but the additional information from the prognostic cloud droplet and ice crystal number concentrations can also improve the simulation of clouds compared to a 1-moment cloud microphysics scheme. The general representation of clouds in ECHAM-HAM is described in the literature given in this section but is briefly repeated in the subsections below for the convenience of the reader.

### 2.1.1 Liquid stratiform clouds

The scheme for stratiform clouds comprises prognostic variables for water vapor, cloud liquid and cloud ice, a cloud microphysics scheme and a diagnostic cloud cover scheme (based on Sundqvist et al., 1989). Cloud microphysics is represented by a 2-moment scheme described in Lohmann et al. (2007), Lohmann and Hoose (2009) and Lohmann and Neubauer (2018). Optionally available but not used in this study is the 1-moment scheme by Lohmann and Roeckner (1996). In ECHAM6.3 changes were made in the diagnostic cloud cover scheme to enhance the cloud cover for marine stratocumulus clouds (Mauritsen et al., 2019). Condensation of cloud liquid water is based on moisture convergence (from transport by advection, turbulence and convection) and subsequent saturation adjustment. Evaporation of cloud liquid water (or sublimation of cloud ice) occurs when the cloud cover decreases, or by transport of cloud liquid (or ice) mass into the cloud free part of a gridbox. For aerosol activation in liquid clouds the Köhler theory-based Abdul-Razzak and Ghan (2000) scheme is used. Its implementation is described in Stier (2016). Optionally available is the Lin and Leaitch (1997) aerosol activation scheme. Precipitation is computed diagnostically. Autoconversion of cloud droplets to rain follows Khairoutdinov and Kogan (2000). Accretion of cloud droplets by rain (Khairoutdinov and Kogan, 2000) and evaporation of rain below



clouds (based on Rotstayn, 1997) are also computed. Size dependent wet scavenging of aerosol particles in-cloud and below cloud follows Croft et al. (2009, 2010). The below cloud collection efficiencies as a function of aerosol and rain drop or snow crystal size are read from a look-up table. The in-cloud scavenging scheme takes nucleation scavenging and impaction scavenging of aerosol particles with cloud droplets and ice-crystals into account. For nucleation scavenging the number of

scavenged aerosol particles is computed for clouds warmer than -35°C from the cloud droplet number concentration (CDNC), the ice crystal number concentration (ICNC) and the number of activated aerosol particles computed by the activation scheme. For clouds colder than -35°C the aerosol particles are scavenged progressively from the largest to the smallest modes.

### 2.1.2 Mixed-phase and cirrus stratiform clouds

The cirrus scheme follows Kärcher and Lohmann (2002) and details are given in Lohmann et al. (2008). The ice crystals in cirrus clouds form by homogenous nucleation of supercooled liquid droplets. The scheme by Joos et al. (2010) for orographic cirrus cloud can optionally be applied to account for the higher updraft velocities in orographic cirrus cloud but was not used in this study. Supersaturation with respect to ice is allowed for cirrus clouds, therefore the depositional growth equation is solved for cirrus ice crystals (Kärcher and Lohmann, 2002). For mixed-phase clouds heterogeneous nucleation of

supercooled cloud droplets is computed via immersion and contact freezing following Lohmann and Diehl (2006). Depositional growth of cloud ice in mixed-phase clouds is computed analogous to liquid clouds based on moisture convergence and subsequent saturation adjustment. In addition, the growth of ice crystals at the expense of cloud droplets via the Wegener-Bergeron-Findeisen process (Wegener 1911; Bergeron 1935; Findeisen 1938) is implemented following Korolev (2007). Snow forms by aggregation of ice crystals, riming of cloud droplets by snow and accretion of ice crystals by

snow. Sedimentation of ice crystals follows Rotstayn (1997). Sublimation and melting of ice crystals and snow below clouds is also computed. Ice multiplication via rime splintering (Hallet-Mossop process) following Levkov et al. (1992) is optional (not used in this study).

### 2.1.3 Convective clouds

The convective parameterization from Tiedtke (1989) with modifications for deep convection from Nordeng (1994) and for

the triggering of convection from Stevens et al. (2013) is used. The convective parameterization uses only a 1-moment cloud microphysics scheme. Detrained condensate of convective clouds is added to stratiform clouds if they exist at the level of detrainment. Whether the condensate is detrained as liquid or ice is based on the same criteria as in the 2-moment stratiform cloud microphysics scheme in ECHAM-HAM. CDNC from convective clouds is weighted by detrained cloud water and CDNC from stratiform clouds is weighted by the stratiform cloud water when adding detrained CDNC to the CDNC of the

stratiform cloud in E63H23 (CDNC of the stratiform cloud is not allowed to decrease by this procedure). The detrained ICNC is computed from the temperature dependent empirical relationship of Boudala et al. (2002). An alternative convection





scheme based on the Convective Cloud Field Model (CCFM) (Wagner and Graf, 2010) with representation of aerosol-convection interactions is available (Kipling et al., 2017; Labbouz et al. 2018) but not used in this study.

### 2.1.4 Other processes

The sulfur cycle model of Feichter et al. (1996) forms the base of the sulfur chemistry module. Three sulfur species are treated prognostically: sulfur dioxide, dimethyl sulfide (DMS) and sulfate (the latter not only in the gas phase but also as an aerosol). Three-dimensional climatological fields for oxidants are used i.e. ozone ($O_3$), OH, $H_2O_2$, $NO_2$ and $NO_3$. The nucleation scheme was implemented by Kazil et al. (2010) and is based on Kazil and Lovejoy (2007). Organic nucleation following Kulmala et al. (2006) or Kuang et al. (2008) can optionally be used. Sea salt, dust and DMS emissions are

computed online based on near surface wind speeds (Stier et al, 2005; Tegen et al., 2002). The Long et al. (2011) sea salt parameterization (temperature dependent; Sofiev et al., 2011) is used in E63H23 and the Guelle et al. (2001) sea salt parameterization is used in E55H20 and E61H22. Aerosol water uptake is computed via kappa-Köhler theory (Petters and Kreidenweis, 2007) as described in Zhang et al. (2012).

Radiative transfer is computed with the two-stream model PSrad (Pincus and Stevens, 2013). Turbulent fluxes in the

atmosphere are computed with the turbulent kinetic energy (TKE) scheme of Brinkop and Roeckner (1995). The subgrid-scale vertical velocity that is needed for many cloud microphysical processes (e.g. cloud droplet activation, ice crystal nucleation, Wegener-Bergeron-Findeisen process) is computed from the TKE (Lohmann et al., 2007). Next to a single characteristic updraft velocity for a gridbox which is based on TKE, there is also the option to represent the subgrid-scale variability of updraft velocities by a Gaussian probability density function (pdf) of updraft velocities (West et al., 2014). The

subgrid-scale variability is again assumed to be due to turbulence and the width of the Gaussian pdf is therefore a function of TKE. The impact of the width of the Gaussian pdf on $ERF_{ari+aci}$ is discussed in West et al. (2014). The pdf-approach by West et al. (2014) is optionally available but not used in this study. The physics part of ECHAM6.3 as well as the 2-moment microphysics scheme for stratiform clouds in E63H23 are now energy conserving.

### 2.1.5 Changes and improvements in E63H23

Changes and improvements in E63H23 are also described in Lohmann and Neubauer (2018) and Tegen et al. (2018) and are repeated here shortly for the convenience of the reader. From ECHAM6.1 to ECHAM6.3 the following improvements were made:

- New PSrad radiation scheme (Pincus and Stevens, 2013), which uses the Monte Carlo independent column approximation for fractional cloudiness and has the option for spectrally sparse but temporally dense calculations

- Update of fractional cloud cover scheme, which improves the low-bias of marine stratocumulus clouds; this is motivated by the difficulty of representing the strong inversions of stratocumulus-topped marine boundary layers in global climate models (Mauritsen et al., 2019)



- Update of land model JSBACH (Reick et al., 2013), which uses a new five layer soil hydrology scheme
- Removal of inconsistencies in the convection scheme, convective detrainment and the vertical diffusion scheme to conserve the atmospheric energy budget

The aerosol microphysics scheme HAM2.3 received the following improvements compared to HAM2.0:

- Update of mineral dust emission parameterization which makes use of a satellite-based source mask for Saharan dust sources (Heinold et al., 2016)
- New sea salt emission parameterization based on Long et al. (2011) which uses a temperature dependence following Sofiev et al. (2011)
- The latest version of the sectional aerosol module SALSA2.0 is implemented (described in Kokkola et al., 2018)

- New emission datasets have been made available in an input file distribution for E63H23 for anthropogenic aerosol emissions

Aerosol-cloud interactions were improved from HAM2.0 to HAM2.3 by the following changes:

- The Köhler-theory based Abdul-Razzak and Ghan (2000) cloud droplet activation scheme (described in Stier, 2016) replaces the empirical Lin and Leaitch (1997) activation scheme

- The in-cloud scavenging scheme by Croft et al. (2010) combines a diagnostic nucleation scavenging scheme with a size-dependent impaction scavenging parameterization and replaces prescribed (size-dependent) aerosol scavenging fractions
- Changed treatment of detrained cloud water mass and number concentrations from convective clouds: CDNC is weighted by detrained cloud water when adding it to the CDNC of the stratiform cloud; split between liquid water

and ice of detrained condensate is made consistent between mass and number concentrations
- In mixed-phase clouds the heterogeneous freezing by immersion freezing of black carbon particles is limited to particles in the accumulation mode and coarse mode

The 2-moment stratiform cloud microphysics scheme in ECHAM-HAM received the following improvements from Lohmann and Hoose (2009) to E63H23:

- Ice crystals are assumed to have a shape of hexagonal plates which covers the whole size range of ice crystals and the shape is consistent in all modules
- Sticking efficiency used in the accretion of ice crystals by snow has been changed to the one used in Seifert and Beheng (2006)
- Two settings for minimum CDNC are available: 40 cm$^{-3}$ or 10 cm$^{-3}$

Further technical improvements, bugfixes and minor corrections in E63H23 include:

- Removal of an inconsistency in the fractional cloud cover and cloud microphysics schemes in ECHAM6.3, which had led cloud cover to be either 0 or 1 in ECHAM6.1
- Removal of inconsistencies in the kappa-Köhler water uptake in HAM2.3



- Modularization of the 2-moment stratiform cloud microphysics scheme
- Removal of an inconsistency for convective detrainment in the 2-moment stratiform cloud microphysics scheme to conserve the atmospheric energy budget
- Removal of an inconsistency in the 2-moment stratiform cloud microphysics scheme which led to homogeneous freezing of dry aerosol particles, independent of availability of water vapor below -35°C
- CDNC/ICNC can no longer grow and in the same timestep evaporate or sublimate
- No more CDNC at temperatures colder than 238.15K, no more ICNC at temperatures warmer than 273.15K
- Update of default settings, run templates and run organization

## 2.2 Experiment description

For each of the three model configurations, E55H20, E61H22 and E63H23 three types of experiments were conducted to evaluate the clouds in present day climate, $ERF_{ari+aci}$ and the equilibrium climate sensitivity (ECS) (Table 1). The simulation setup was chosen to be as similar as possible for the three model versions to minimize the impact of boundary conditions on the model version comparison. The setup represents the standard setup of E63H23 which is a compromise between which processes are represented in the model and the computational performance. Climatological monthly mean mixing ratios of oxidants from an eight-years (2003-2010) mean Monitoring Atmospheric Composition and Climate (MACC) reanalysis (Inness et al., 2013) are used in E61H22 and E63H23. For E55H20 the climatological monthly mean mixing ratios of oxidants are from simulations with the MOZART model for present day conditions (Horowitz et al., 2013).

### 2.2.1 Present day climate simulation (PD)

10-year simulations for PD conditions were done for all model versions. Previous studies using E55H20 or E61H22 often use the year 2000 as the reference year or 2000-2009 as the reference period for present day simulations (Zhang et al., 2012; Neubauer et al., 2014), therefore we also use the period 2000-2009 for the PD simulations of E55H20 and E61H22. For E63H23 the default model setup has changed and the reference year and reference period for present day simulations are now 2008 and 2003-2012 respectively. This has become necessary because of the relatively large changes in aerosol emissions in recent years in several regions (Hoesly et al., 2018) and was aided by the availability of new boundary condition datasets for the Coupled Model Intercomparison Project (CMIP) Phase 6 (CMIP6). Time varying (RCP8.5) ACCMIP MACCity (AEROCOM II ACCMIP) aerosol emissions were used for E63H23 and E61H22. The biomass burning emissions are based on observations until 2008 in ACCMIP MACCity, afterwards the biomass burning emissions are from the RCP8.5 emission scenario. For E55H20 the AEROCOM I emission for the year 2000 are applied for all years. The greenhouse gas concentrations are set to year 2008 (RCP8.5) concentrations in all model versions. All model versions also use a climatology for monthly values of sea surface temperature (SST) and sea ice cover (SIC) derived from AMIP data (Taylor et al., 2000) of the years 2000-2015. The spectral horizontal resolution is T63 (~1.9°x1.9°) in all model versions. For



E55H20 and E61H22 31 vertical model layers (L31) are used (as in the default configuration of these model versions), the model top is 10 hPa. For E63H23 47 vertical model layers (L47) are used (new default configuration of E63H23), with a model top at 0.01 hPa. The lowermost levels of L31 and L47 (up to about 100 hPa) are identical. A comparison of E63H23 simulations with L31 and L47 showed only minor differences (see Table S1 and Fig. S1). Therefore, we expect also no large
differences by using different vertical grids for the different model versions.

### 2.2.2 ERF$_{ari+aci}$ and RF$_{ari}$ simulations (PD$_{aer}$/PI$_{aer}$)

To compute ERF$_{ari+aci}$ two 20-year simulations were conducted, one with present day aerosol emissions (PD$_{aer}$) and one with pre-industrial aerosol emissions (PI$_{aer}$). The simulations were otherwise identical. For the PD$_{aer}$ simulation the PD simulation was extended to cover the years 2000-2002 and 2013-2019 (or the years 2010-2019 for E55H20 and E61H22). The same
greenhouse gas concentrations and SST and SIC climatology as in the PD simulations have been used. For the time period 2013-2019 the ACCMIP MACCity (AEROCOM-II-ACCMIP) aerosol emissions for the year 2008 were used for all years for E63H23 and E61H22. For E55H20 the AEROCOM I emission for the year 2000 are applied for all years (2000-2019) for the PD simulation. For the PI$_{aer}$ simulation the aerosol emission for the year 1850 from ACCMIP MACCity (AEROCOM-II-ACCMIP) were used for E63H23 and E61H22 and the ones for the year 1750 from AEROCOM I for E55H20. ERF$_{ari+aci}$ is
computed as the difference in top of atmosphere (TOA) net radiative flux ($net^{TOA}$) between the PD$_{aer}$ and the PI$_{aer}$ simulation:

$$ERF_{ari+aci} = net^{TOA}_{PD_{aer}} - net^{TOA}_{PI_{aer}}. \qquad (1)$$

The simulation time was 20 years to increase the signal of ERF$_{ari+aci}$ compared to variations in TOA net radiative flux due to internal variability of the climate system.
The radiative forcing due to aerosol-radiation interactions (RF$_{ari}$) is computed from the same pair of simulations as ERF$_{ari+aci}$ (PD$_{aer}$/PI$_{aer}$). The direct aerosol radiative effect is computed by double calls to the radiation, once with the prognostic aerosol and once without aerosol. RF$_{ari}$ is computed as the difference of the direct aerosol radiative effect between the PD$_{aer}$ and the PI$_{aer}$ simulations at TOA, once for all-sky and once for clear-sky (CS) conditions:

$$RF_{ari} = \left(net^{TOA} - net^{TOA}_{no\_aer}\right)_{PD_{aer}} - \left(net^{TOA} - net^{TOA}_{no\_aer}\right)_{PI_{aer}}, \qquad (2a)$$

$$RF_{ari,CS} = \left(net^{TOA,CS} - net^{TOA,CS}_{no\_aer}\right)_{PD_{aer}} - \left(net^{TOA,CS} - net^{TOA,CS}_{no\_aer}\right)_{PI_{aer}}. \qquad (2b)$$

Note that we did not follow the protocol in Myhre et al. (2013) for all-sky conditions, therefore our all-sky RF$_{ari}$ is somewhat affected by changes in clouds from pre-industrial to present day simulations. This has no large impact on the regional analysis for RF$_{ari}$ our study. The reason why we did not follow Myhre et al. (2013) is that we include indirect aerosol effects
in our simulations.



### 2.2.3 ECS simulations (1xCO$_2$/2xCO$_2$)

To compute ECS, ECHAM-HAM was coupled to a mixed-layer ocean to compute two 50-year simulations, one with pre-industrial CO$_2$ concentrations (1xCO$_2$) and one with doubled pre-industrial CO$_2$ concentrations (2xCO$_2$). The first 25 years of the simulations were used as spin-up time for the (50 m deep) mixed-layer ocean. ECS was then computed from the difference in global mean surface temperature ($T_s$) between 2xCO$_2$ and 1xCO$_2$ from the last 25 years of the simulations:

$$ECS = T_s^{2 \times CO_2} - T_s^{1 \times CO_2}. \tag{3}$$

Pre-industrial concentrations for well-mixed greenhouse gases other than CO$_2$ were used in all simulations as well as pre-industrial aerosol emissions (similar to PI$_{aer}$). The ocean heat flux corrections required by the mixed-layer ocean to maintain present-day sea surface temperatures were computed for each model version by extending the respective PD$_{aer}$ simulations another 5 years to a total of 25 years.

### 2.3 Tuning strategy

Following Hourdin et al. (2017) who argue that estimating uncertain parameters in model development is an important process that should be made transparent we document here our tuning strategies and targets. Tuning is needed mainly to ensure that the TOA radiative fluxes are balanced, and model tuning is limited to adjusting global mean properties. We start from the ECHAM6.3 parameter settings and adapt mainly parameters related to the cloud and convection scheme for tuning E63H23. The tuning strategy and parameters for ECHAM6 as well as the impact of these parameters on the model climate are described in Mauritsen et al. (2012). The tuning parameters for ECHAM6-HAM2 and their impact on climate are described in Lohmann and Ferrachat (2010). The parameters that were used in the tuning of the ECHAM-HAM versions and that have different values in E55H20, E61H22 and E63H23 are shown in Table 2.

The primary tuning target for E63H23 is to match the global mean observed shortwave (SW) and longwave (LW) TOA fluxes within the range of uncertainty of the observations and that the net radiative imbalance TOA is close to the observed present day value. The secondary tuning target is that the SW, LW and net cloud radiative effect (CRE) TOA are within the range of uncertainty of the observations. Also cloud cover (CC), liquid water path (LWP), ice water path (IWP), total precipitation (P) and aerosol optical depth (AOD) should be close to the range of observed values (see Table 3).

The tuning is done with short one-year simulations with a climatology for SST and sea ice. When a set of parameters has been found one or more ten-year simulations are done to minimize the uncertainty in net radiative imbalance TOA. For E61H22 the default parameter values are used (Neubauer et al., 2014). For E55H20 it was necessary to retune the model with the tuning strategy described above as the tuning in Zhang et al. (2012) was undertaken for nudged simulations and we performed free simulations to compare the three ECHAM-HAM model versions. The largest differences in tuning between the three model versions are in the tuning parameters for autoconversion of cloud droplets to rain and entrainment for shallow convection. In E63H23 the stratiform rain formation by autoconversion will be faster than in the other two model



versions. This is due to the larger value of the respective tuning parameter leading to reduced LWP, CC, SW CRE and more positive net radiative imbalance TOA in E63H23 (Lohmann and Ferrachat, 2010). The larger value of the tuning parameter for entrainment for shallow convection in E61H22 and the even larger value in E63H23 have the opposite effect, increased LWP, CC, SW CRE and more negative net radiative imbalance TOA (Mauritsen et al., 2012). For E63H23 there is a compensation by changing both tuning parameters, the most pronounced net effect is a reduced LWP compared to the two other model versions (we hypothesize that LWP is reduced since entrainment for shallow convection affects mainly low, thin clouds, whereas the autoconversion rate affects all liquid clouds; since the reflectivity of clouds depends non-linearly on their thickness, an increase in thin low clouds can compensate the SW CRE change by the decrease in thicker clouds but lead to a lower global mean LWP).

## 2.4 Observational products

We list here the products and the respective references for the observational products used in the model evaluation. From Moderate-resolution Imaging Spectroradiometer (MODIS-AQUA) collection 6.1 (Platnick et al., 2015, 2017) and from the ESA Cloud Climate Change Initiative (CCI)-Advanced Very-High-Resolution Radiometer (AVHRR-PM) v3.0 (prototype; Stengel et al., 2017a, 2017b), histograms of cloud top pressure vs. cloud optical depth and CC are taken and from MODIS also LWP. Histograms of cloud top pressure vs. cloud optical depth were also taken from the International Satellite Cloud Climatology Project (ISCCP; Rossow and Schiffer, 1999) D1 data. Cloud-Aerosol Lidar and Infrared Pathfinder Satellite Observations (CALIPSO) data for CC is from the GCM-Oriented CALIPSO Cloud Product (GOCCP) dataset (Chepfer et al., 2010). Cloud radiative effect data is from the Clouds and the Earth's Radiant Energy System (CERES) Energy Balanced and Filled (EBAF) TOA edition-4.0 data product (Loeb et al., 2018). Precipitation data is from the Global Precipitation Climatology Product (GPCP) 2.3 (Adler et al., 2018). Cloud top CDNC are from the climatology of Bennartz and Rausch (2017). LWP data are from the Multi-Sensor Advanced Climatology of LWP (MAC-LWP; Elsaesser et al., 2017), an updated version of the University of Wisconsin LWP climatology. IWP is from satellite observations compiled by Li et al. (2012).

## 3 Results and Discussion

### 3.1 Global mean comparison to observations

Table 3 includes global mean values of radiation, cloud and aerosol related variables of the PD simulations of E55H20, E61H22 and E63H23 compared to observations (OBS) or multi-model mean values (MMM) when observations are not available. The global mean values of the radiative fluxes shown in Table 3 are tuning targets (see section 2.6) and therefore cannot be used directly for model evaluation. For E63H23 the SW and LW TOA fluxes as well as the SW and LW CRE TOA are within the range of the observations. The net TOA flux of E63H23 is also close to the observations (additional tuning could bring it closer to the observed value but was not attempted given the large uncertainty in e.g. SW and LW TOA



fluxes). The SW, LW and net TOA fluxes of E61H22 and E55H20 are outside the range of observations. This reflects the change in the tuning targets/strategy in E63H23 and the availability of better observations. The net CRE of E63H23 (and also E55H20 and E61H22) is outside the observed range. It was not possible to find parameter settings which bring the net CRE within the range of observations without bringing one or more of the other radiative fluxes outside of the range of observations. This is a first indication of a structural problem in ECHAM-HAM, which could be related to how ice crystals nucleate in (warming) cirrus clouds or an underestimation of (cooling) stratocumulus. This will be further discussed in the evaluation. CC, P and cloud top cloud droplet number concentration (CDNC) of all three model versions agree fairly well with observations (for cloud-top CDNC of the model simulations we selected CDNC over ocean only of the topmost layer of clouds with a cloud top temperature > 273.15 K). For LWP a climatology based on microwave sensors (Elsaesser et al., 2017) provides reliable observations as long as the ratio of LWP to LWP+rain water path is large (>0.8 is used here), i.e. in regions with relatively low precipitation. The values of LWP only in this low precipitation regions (LWP-LP) are also shown in Table 3. Whereas the mean values over the global oceans for LWP and LWP-LP of E61H20 and E55H20 are higher than observed, E63H23 shows values within the observational range (71 and 76 g m$^{-2}$ respectively). This is due to the more physically based activation scheme in E63H23 and improvements in ECHAM6.3 like energy conservation in the physics part and improvements in the cloud cover scheme for marine stratocumulus clouds, which allow to increase the tuning parameter for autoconversion (see Table 2). Similarly, the global mean value of IWP in E63H23 with 15 g m$^{-2}$ is only slightly below the observational range (18-32 g m$^{-2}$), whereas in E61H22 and E55H20 IWP is considerably smaller (10 and 8 g m$^{-2}$ respectively). This is because in the accretion of ice crystals by snow, the sticking efficiency follows Seifert and Beheng (2006) in E63H23, whereas in E55H20 and E61H22 it followed Levkov et al. (2012). The Seifert and Beheng (2006) sticking efficiency leads to a less efficient removal of ice crystals by snow. Furthermore, the changed sticking efficiency allows to reduce the stratiform snow formation rate by aggregation compared to earlier model versions (see Table 2), which also increases IWP. The aerosol mass burdens of the five prognostic aerosol species in ECHAM-HAM are within the range of AeroCom models (Textor et al., 2006), except for particulate organic matter (POM). This may be related to the simplistic treatment of secondary organic aerosol (SOA) in all three model versions in the experiments for this study. Details of the evaluation of E63H23 with respect to the atmospheric aerosol are given in Tegen et al. (2018).

## 3.2 Zonal mean comparison to observations

Although the global mean values are tuning targets (see section 2.6), biases in net CRE and IWP in the ECHAM-HAM versions, which could not be brought in agreement with observations via tuning, were identified in the previous section. Zonal mean values of observable variables can nevertheless be used for model evaluation because tuning targets the global mean quantities. Fig.1 shows zonal mean distributions of several quantities for the three model versions and observations. The zonal distribution of SW CRE and LWP-LP of E63H23 agree relatively well with observations, whereas in E61H22 and E55H20 the magnitude of both quantities is overestimated in mid-latitudes. The cloud over distribution of E63H23 also agrees well with observations, whereas E61H22 and E55H20 show an underestimation by up to 10 percentage points in the



subtropics. Biases in cloud top CDNC are more complex and retrievals of cloud top CDNC are only possible for specific clouds (e.g. horizontally homogeneous, unobscured, optically thick clouds) and rely on assumptions like that the liquid water content increases with altitude like in an adiabatically rising cloud parcel (or at least like a constant fraction of this liquid water content) and that CDNC are constant throughout the cloud and further assumptions which together make cloud top

CDNC retrievals uncertain (Grosvenor et al., 2018). We therefore expect larger differences between observations and models for cloud top CDNC than for other variables. E55H20 agrees well with MODIS observations in the tropics but overestimates cloud top CDNC in the subtropics on both hemispheres and mid latitudes in the Southern Hemisphere. E61H22 overestimates cloud top CDNC in the tropics and subtropics but underestimates it at mid latitudes in the Northern Hemisphere. E63H23 also overestimates cloud top CDNC in the subtropics, however less than E61H22, and also in the

tropics. The liquid phase of clouds is therefore better represented in E63H23 than in the previous model versions. IWP is underestimated in all three model versions. E63H23 has the smallest bias, followed by E61H22 and E55H20 shows the largest deviation from observed zonal mean IWP. The underestimation is particularly large in the tropics. For LW CRE (and precipitation and AOD, not shown) all three model versions differences are within the range of different observational products. In the tropics E55H20 has a rather strong LW CRE, whereas E63H23 and E61H22 have a rather weak LW CRE

but all are within the range of observations.

### 3.3 Regional comparison to observations

The comparison of CRE of the different model versions with CERES CRE reveals several biases in the representation of clouds. We start therefore by identifying biases in CRE and use then observations for other quantities to identify causes of the model biases. In Fig. 2 the differences in SW, LW and net TOA CRE of all model versions to CERES observations are

shown. In all three model versions the (negative) SW CRE is too weak in the marine stratocumulus regions west of the continents (the average bias in the wider stratocumulus regions is 1.1, 8.1 and 7.0 W m$^{-2}$ in E63H23, E61H22 and E55H20 respectively). In addition the SW CRE is too weak in some land areas in E63H23 and E61H22, in the Southern Ocean in E63H23 and in the tropical oceans in E55H20 (3.3 W m$^{-2}$ average bias over ocean between 15°N and 15°S excluding wider stratocumulus regions). These biases are compensated by a stronger SW CRE over large parts of the oceans and mid and

high latitude land areas in the Northern Hemisphere. The bias in stratocumulus regions is smaller in E63H23 than in the older model versions and so are the compensating negative biases. This is due to improvements in ECHAM6.3 like improvements in the cloud cover scheme for marine stratocumulus clouds. The (positive) LW CRE is too weak in the tropics in E63H23 and E61H22 (-7.7 and -4.8 W m$^{-2}$ average bias respectively between 20°N and 20°S excluding wider stratocumulus regions) and too strong in the tropics (in particular over land) in E55H20 (1.2 W m$^{-2}$). Together with the

biases in the tropical SW CRE this points to problems with the parameterization of deep convective clouds, detrained condensate or the representation of anvils from detrained condensate in all three model versions. In all three model versions the LW CRE in mid-latitudes is too weak (except over land in the Northern Hemisphere). At high latitudes it is stronger than in the CERES data in all model versions (but also the uncertainty of CERES CRE is larger at high latitudes, Loeb et al.,



2018). Only few of the biases in SW and LW CRE compensate, therefore the biases in net CRE are as large as or larger than in the SW (LW) CRE. In the net CRE the positive biases in stratocumulus regions as well as in the Southern Ocean in E63H23 and E61H22, over land in E61H22 and in the tropical oceans in E55H20 are compensated in the global mean by negative biases in all other regions. The negative biases are caused by adjusting cloud parameters to bring the global mean

values in agreement with observations. Therefore, if the biases in stratocumulus regions and the Southern Ocean (and the tropics) could be reduced, the negative biases in SW and net CRE would also be smaller.

In Fig. 3 the cloud cover of the CALIPSO GOCCP product and all three model versions are shown. The hatched areas in Fig. 3 are the regions where the cloud cover of CALIPSO GOCCP, MODIS collection 6.1 and ESA Cloud CCI (AVHRR-PM) differ by more than five percentage points. We use therefore only the areas not marked by hatching in Fig. 3 for the model

evaluation. The cloud cover of all three model versions agrees fairly well with the observations. The largest biases are in stratocumulus regions west of the continents (-10, -18 and -22 percentage points in E63H23, E61H22 and E55H20 respectively in the wider stratocumulus regions), where the models underestimate the cloud cover. Over land in the Northern Hemisphere the models overestimate cloud cover and in the Indonesian warm pool region the cloud cover is biased high in the three model versions. The underestimation of cloud cover in stratocumulus regions is less severe in E63H23 than in the

other two model versions (the cloud cover scheme in ECHAM6.3 was improved to better represent cloud cover in these regions).

Fig. 4 shows LWP from the MAC-LWP climatology (Elsaesser et al., 2017) and the three model versions. The retrieval of LWP has biases both from visible/near infrared sensors as well as microwave sensors (Seethala and Horváth, 2010; Lebsock and Su, 2014). Visible/near infrared sensors such as MODIS have problems when the solar zenith angle is large or at

detecting pixels of clouds at low altitudes (Lebsock and Su, 2014). Retrievals of microwave sensors such as AMSR-E may retrieve LWP in cloud free scenes and the split between LWP and rain water path is difficult (Lebsock and Su, 2014). Elsaesser et al. (2017) corrected the retrieval bias of LWP of microwave sensor based products in cloud free scenes. And they recommend to use regions with low precipitation (LWP/(LWP+rain water path)>0.8) for model evaluation. The regions where precipitation could influence the LWP retrieval are therefore hatched in Fig. 4. This leaves the stratocumulus regions

west of the continents and the storm tracks over ocean in the Northern and Southern Hemisphere as the most reliable areas for the evaluation of LWP. All three model regions show a fairly good agreement of LWP in the stratocumulus regions except west off South America and southwest Africa, where all model versions tend to underestimate LWP. In the storm tracks over ocean in the Northern and Southern Hemisphere on the other hand E61H22 and even more E55H20 overestimate LWP. E63H23 shows instead a rather good agreement of LWP in the storm tracks compared to observations. This is likely

the result of different model tuning in E63H23 (see section 2.3), which was possible due to a more realistic geographic pattern of cloud cover and SW CRE in E63H23.

To further characterize the simulation of liquid clouds in the ECHAM-HAM model versions we also compare cloud top CDNC of warm (cloud top warmer than 0°C) liquid clouds to the cloud top CDNC from Bennartz and Rausch (2017), which is based on MODIS Aqua data (Fig. 5). The hatched area marks regions where the relative uncertainty in the observations is



larger than 75%. The general geographical distribution and magnitude of cloud top CDNC of all model versions agree with the observations, although there are certain areas where model biases are apparent. E61H22 has lower cloud top CDNC concentrations in mid latitude ocean regions than E63H23, in E55H20 they are higher than in the other two model versions. In E55H20 the higher cloud top CDNC concentrations can be explained by reduced entrainment of deep convection (see

Table 2) compared to the other model versions which leads to higher relative humidity in the upper tropical troposphere which subsequently leads to increased aerosol nucleation, more Aitken mode particles and increased cloud condensation nuclei (CCN) concentrations. E63H23 uses the Abdul-Razzak and Ghan (2000) activation scheme and the Long et al. (2011) sea salt emission parameterization (temperature dependent; Sofiev et al., 2011) which lead to higher cloud top CDNC concentrations than in E61H22 (which uses the Lin and Leaitch, 1997 activation scheme and the Guelle et al, 2001 sea salt

emission parameterization) despite the lower LWP in E63H23. Furthermore, in subtropical regions where the cloud cover and LWP are low (see Figs. 3 and 4), cloud top CDNC are higher in all three model versions than in the observations. In these regions, shallow trade-wind cumulus clouds occur frequently (Medeiros and Stevens, 2011) and in all model version shallow convection is triggered frequently (see Fig. S2). For convective clouds a 1-moment cloud microphysics scheme is used in ECHAM-HAM and these clouds are considered to be short-lived. But the detrained condensate of these clouds can

add to existing large-scale clouds. To obtain CDNC for the detrained condensate, the number of activated CCN at the convective cloud base (computed using the vertical velocity from large scale and turbulent fluxes as described in section 2.1.4) is propagated as CDNC to the level of detrainment. The detrained CDNC are then added to an existing stratiform cloud (otherwise they dissipate) by taking an average of the stratiform CDNC and detrained CDNC (weighted by their respective LWP). This weighted average may overestimate the CDNC concentration of shallow cumulus clouds. The use of

a 2-moment cloud microphysics scheme for convective clouds (e.g. Lohmann, 2008) so that CDNC in convective clouds can be reduced by collision-coalescence or a different way to account for detrained CDNC could help to alleviate this model bias. All three model versions also underestimate cloud top CDNC at high latitudes. As retrievals of visible/near infrared sensor often have biases at large zenith angles (see above) this may be a problem with the observations.

In Fig. 6 the IWP of all three model versions and IWP satellite observations compiled by Li et al. (2012) are shown. Li et al.

(2012) used three different CALIPSO plus CloudSat ice water products and two different methods to remove the contribution of convective clouds and precipitation from the products. Fig. 6 displays the compiled mean IWP of the datasets of Li et al. (2012) and areas where the relative standard deviation of the different datasets is larger than 75% are hatched. The regional distribution of IWP of all three model versions agrees in general quite well with the observations. IWP is biased low in all model versions, as was found for the respective global mean and zonal mean values (see sections 3.1 and 3.2).

Similar to what was found in the analysis of zonal mean IWP the underestimation is largest in the tropics.

Cloud ice mass vertical profiles can be obtained from CALIPSO plus CloudSat observations. The global mean vertical profile of IWC is shown in Fig. 7 for all three model versions and the compiled mean IWC from Li et al. (2012). IWC is underestimated above 700 hPa in all model versions. In E63H23 the maximum of IWC is at the same pressure level as in the observations, at about 350 hPa, whereas in E61H22 and E55H20 the maximum of IWC is at higher altitudes at about 300 to





250 hPa. This can be explained by changes in ICNC and subsequent changes in precipitation formation and ice crystal sedimentation. ICNC changed between the model versions because of the way detrained ice crystals are added to existing stratiform clouds was changed since E61H22. The shape of the ice crystals was made consistent in all modules since E61H22 and a bug in E61H22 was removed in E63H23 which led to homogeneous freezing of dry aerosol particles,

independent of availability of water vapor below -35°C (the latter improvement was most important as it doubled the ICNC burden in E61H22 compared to the two other model version; see Table 3). Below 700 hPa the three model versions are close to the observed IWC, with E63H23 showing an overestimation of IWC and E55H20 an underestimation.

The regions where IWP is underestimated in the three model versions correspond to the regions where the three model versions underestimated LW CRE in Fig. 2 (in particular in the tropics). There are also regions where LW CRE is

overestimated (see Fig. 2) in the three model versions although IWP is underestimated (see Fig. 6). This is an indication that ICNC is too large in the three model versions (the vertical profile of IWC agrees fairly well with observations although the IWC magnitude is underestimated in all three model versions). As IWP is larger in E63H23 than in E61H22 and E55H20 but the overestimation in LW CRE is smaller in E63H23 than in E61H22 and E55H20, this is an indication that ICNC and the size of the ice crystals are closer to reality in E63H23 than in E61H22 and E55H20. The overestimation of LW CRE in

E61H22 around 60°N and 60°S can be explained by the high ICNC in E61H22 (see Table 3) caused by the ICNC bug mentioned above.

Next to E61H22 there is also a bias of net CRE south of 60°S in E63H23 (see Fig. 2). This is not due to too high ICNC as LW CRE of E63H23 agrees well with CERES observations in this region. In E63H23 there is an underestimation of SW CRE south of 60°S. Cloud cover and IWP of E63H23 agree well with observations in this region. LWP is slightly

underestimated, but cloud top CDNC is strongly underestimated. Either there is a problem with the satellite retrievals at these high southern latitudes or E63H23 is missing liquid clouds in this region. In E61H22 and E55H20 this possible bias south of 60°S is hidden by the overestimation in LWP which leads to a stronger SW CRE.

In Fig. 8 the total precipitation of all model versions and GPCP2.3 (Adler et al., 2018) is shown. Areas where the relative uncertainty of the GPCP2.3 data is larger than 75% are hatched. Despite the biases in the representation of clouds in the

three model versions identified above, the geographical distribution and magnitude of the annual mean precipitation of all model versions agrees well with the observations. Only in the intertropical convergence zone (ITCZ) and South Pacific convergence zone (SPCZ) the areas and magnitude of precipitation differs from the observations, corresponding to differences in cloud cover and IWP (Fig. 3 and Fig. 6, respectively). Cloud cover, IWP and precipitation are low in the central Pacific and central Atlantic ITCZ but relatively large in the ITCZ west off Central America, east off South America,

over the Philippines and west off Southeast Asia. In the SPCZ cloud cover, IWP and precipitation are relatively large compared to the respective observations.

In Fig. 9 histograms of cloud top pressure vs. cloud optical depth of ECHAM-HAM are compared to ISCCP, AVHRR-PM and MODIS observations. The COSP simulator (Bodas-Salcedo et al., 2011) was not implemented in E55H20 so we only compare E61H22 and E63H23 to the observations. We applied the COSP-ISCCP-simulator for E61H22 and E63H23 for



comparison to ISCCP and AVHRR-PM. The COSP-MODIS-simulator is only implemented in E63H23 so we compare only E63H23 to MODIS. The histograms were produced for four regions (shown in Fig. 2): wider stratocumulus regions, mid-latitudes, tropics and 60°N to 60°S. The five marine stratocumulus regions are west off North and South America, west off northern and southern Africa and west off Australia. The marine stratocumulus regions are extended to the west to cover

approximately the regions were the three model versions underestimate SW CRE (see Fig. 2). The mid-latitudes regions are 60°N to 20°N and 20°S to 60°S, excluding the areas covered by the wider stratocumulus regions. The tropics are between 20°N to 20°S, excluding the areas covered by the wider stratocumulus regions. The region 60°N to 60°S is the sum of the wider stratocumulus, mid-latitudes and tropics. 60°N to 60°S was chosen because retrievals of visible/near infrared sensors often have biases at large zenith angles (see above). Several of the biases described above are also seen in the histograms in

Fig. 9. In the region 60°N to 60°S E63H23 and E61H22 simulate too many optically thick clouds and too few optically thin clouds at low and mid-levels compared to the three satellite datasets. Stevens et al. (2013) found a similar bias in ECHAM6.1 ("too few, too bright"). This bias can also be seen in the mid-latitudes and in the tropics. In the wider stratocumulus region on the other hand the occurrence of low-level optically thick clouds of E61H22 and E63H23 agrees rather well with those of the three observational datasets. In the wider stratocumulus regions the optically thin low level (and

mid-level) clouds are missing. This agrees with the analysis of SW CRE that the underestimation in the stratocumulus regions is compensated by a stronger SW CRE (clouds being optically thicker) in other regions (by model tuning). Therefore, if the bias in stratocumulus regions could be reduced, also the biases in SW CRE and cloud optical depth in other regions could be reduced. In the mid-latitudes the optical depth of low, mid- and high level clouds is larger in E61H22 than in E63H23 and ISCCP and AVHRR-PM. This is related to the stronger compensation by tuning for the lack of clouds in

stratocumulus regions (removal of LWP by autoconversion is slower in E61H22 than in E63H23; see Table 2), and to the ICNC bug in E61H22 mentioned above. In the mid-latitudes and the tropics there is also a lack of high level clouds with optical depth between 1.3 and 23 in E63H23 and E61H22. This lack of cirrus clouds corresponds to the underestimation of IWP and LW CRE. Gasparini et al. (2018) evaluated the cirrus clouds in a version of E61H22 that included a cirrus cloud scheme which accounts for a competition in cirrus cloud formation by homogeneous nucleation of solution droplets,

heterogeneous freezing of ice nucleating particles and water vapor deposition on pre-existing ice crystals (Kuebbeler et al, 2014). With this cirrus scheme E61H22 could be tuned such that the global mean IWP agrees with the observations compiled by Li et al. (2012). Similarly Lohmann and Neubauer (2018) made an experiment where cirrus clouds could only form by heterogeneous freezing of ice nucleating particles in E63H23. In their experiment this caused the global mean IWP to agree with the observations compiled by Li et al. (2012). These studies and our analysis indicate that the IWP bias in the three

model versions occurs because cirrus clouds can only nucleate homogeneously and therefore ICNC in cirrus clouds and hence their optical properties are misrepresented.



### 3.4 Summary of model evaluation

Fig.10 shows a Taylor diagram (Taylor, 2001) comparing SW and LW CRE, cloud cover, LWP-LP, cloud top CDNC, IWP and precipitation of the three model versions to the respective observations. The standardized deviations of cloud top CDNC had to be scaled by a factor of 1/3 so they could fit on the scale. For all variables except IWP and precipitation, the root-
mean-square (RMS) error (solid circles in the diagram in Fig. 10) is smaller or equal in E63H23 compared to in E61H22 and E55H20. For IWP, the RMS of E55H20 is smaller than for E63H23 but the bias in global mean IWP is much larger in E55H20 than in E63H23. The changes in the geographical pattern between the three model versions are rather small. E63H23 has somewhat higher correlations except for LW CRE, IWP and precipitation where E55H20 has higher correlations than E63H23 and E61H22. E55H20 has higher correlations of LW CRE, IWP and precipitation because the ratio
of the peaks in these variables in the tropics compared to mid-latitudes is better represented in E55H20 (see Fig.1). Overall E63H23 is an improvement over earlier model versions.

Several biases in the representation of clouds in the three ECHAM-HAM model versions could be identified. The common problem of global climate models (GCMs) in their representation of convective and boundary layer clouds is also present in the three ECHAM-HAM model versions. Stratocumulus clouds are underestimated in all three model versions. Shallow
convective clouds are underestimated in E61H22 and E55H20. In E63H23 the cloud cover and LWP in regions where shallow convective clouds are common agree well with observations but the cloud top CDNC are overestimated, leading to an overestimation of SW CRE in these regions. Deep convective clouds in the Atlantic and Pacific oceans form too close to the continents (see Figs., 3, 6 and 8). IWP is underestimated in all three model versions, in particular in the tropics, whereas LW CRE and the vertical profile of cloud ice agree rather well with observations. This indicates that ICNC may be
overestimated in all three model versions (since LW CRE depends on the cloud temperature ($\sim$ cloud altitude) and cloud optical depth ($\propto$ ICNC, IWP)). As E63H23 has the smallest bias in IWP also the bias in ICNC should be smaller than in the previous versions of ECHAM-HAM. Previous studies (Gasparini et al., 2018; Lohmann and Neubauer, 2018) showed that this overestimation of ICNC is (at least partly) due to missing processes in the formation of cirrus clouds (heterogeneous freezing of ice nucleating particles and/or water vapor deposition on pre-existing ice crystals). South of 60°S LWP and cloud
top CDNC of E63H23 could be underestimated, although there could also be problems with the satellite retrievals at these high latitudes. In the previous model versions this possible bias was hidden by the overestimation of LWP.

### 3.5 Simulation of ERF$_{ari+aci}$, RF$_{ari}$ and ECS

In Fig. 11 global maps of SW and LW ERF$_{ari+aci}$ for E63H23, E61H22 and E55H20 are shown. An important difference exists in the setup of E55H20 and the two other model versions. E55H20 uses AeroCom I aerosol emission data and the
years 1750 and 2000 for pre-industrial and present day aerosol emissions. E63H23 and E61H22 on the other hand use AeroCom II aerosol emissions and the years 1850 and 2008 for pre-industrial and present day aerosol emissions. The stronger SW ERF$_{ari+aci}$ in the east of North America and Europe and the weaker SW ERF$_{ari+aci}$ in South and East Asia in





E55H20 compared to the two other model versions are therefore predominantly the result of the different representative emission years and inventories (see supplementary Fig. S3). We keep these emissions years as they were used as the default in previous studies (e.g. Zhang et al., 2012; Neubauer et al., 2014).

The treatment of surface albedo over land, ocean and sea ice has changed substantially from ECHAM5 to ECHAM6 (see

Stevens et al., 2013), which has an impact on SW $ERF_{ari+aci}$ (Stier et al., 2013). Because of the differences in setup and surface albedo treatment of E55H20 we focus here the comparison of $ERF_{ari+aci}$ on differences between E61H22 and E63H23. $ERF_{ari+aci}$ is stronger over land and weaker over oceans in E63H23 compared to E61H22 (Fig. S4). In the global mean 50% (-0.5 $Wm^{-2}$) of $ERF_{ari+aci}$ originate over land in E63H23 and 50% (-0.5 $Wm^{-2}$) over ocean. This is in contrast to E61H22 where 27% (-0.3 $Wm^{-2}$) of $ERF_{ari+aci}$ originate over land and 73% (-0.9 $Wm^{-2}$) over ocean. The increase in $ERF_{ari+aci}$

in E63H23 over land may be related to the higher rate of autoconversion in E63H23 (Table 2). While Lohmann and Ferrachat (2010) found no strong dependence of $ERF_{ari+aci}$ (or a small decrease) to the autoconversion tuning parameters in the global mean, the ratio of autoconversion to the total rain formation rate has a strong regional dependence (see e.g. Sant et al., 2015) which could lead to regional differences in $ERF_{ari+aci}$.

It is interesting to note that although biases in the simulation of clouds in stratocumulus regions are reduced in E63H23,

there seems to be no increase in $ERF_{ari+aci}$ in these regions compared to E61H22. Over the remote oceans, the largest differences in $ERF_{ari+aci}$ between E63H23 and E61H22 occur between 15°N and 45°N, where E61H22 simulates a strong $ERF_{ari+aci}$ in tradewind cumulus clouds (Zhang et al., 2016). $ERF_{ari+aci}$ in these shallow convective clouds regions is weaker in E63H23, although more clouds are simulated in E63H23 in these regions. LWP and cloud cover in E63H23 are closer to observations in these regions (Figs. 3 and 4), and cloud top CDNC are rather too high in E63H23 (Fig. 5). A smaller LWP

could lead to a weaker $ERF_{ari+aci}$ (Lohmann and Ferrachat, 2010). To better understand the differences over oceans between E63H23 and E61H22, we compare E63H23 also with a simulation with E63H23 where CDNCmin was lowered from 40 $cm^{-3}$ to 10 $cm^{-3}$ (E63H23-10CC) as this simulation has a higher LWP (due to a smaller $\gamma_r$; not shown). Although the smaller CDNCmin of 10 $cm^{-3}$ leads to a stronger $ERF_{ari+aci}$ everywhere (Hoose et al., 2009; -1.7 $Wm^{-2}$ in the global mean; Table S1), this simulation still provides useful information. In the Northern Hemisphere Pacific there is an increase in $ERF_{ari+aci}$ in the

E63H23-10CC compared to E63H23 (Fig. S5). This may be due to the larger LWP or the change in CDNCmin itself. In the Northern Hemisphere Atlantic however, $ERF_{ari+aci}$ does not increase. The weaker $ERF_{ari+aci}$ in the Northern Hemisphere Pacific in E63H23 could therefore be due to the smaller LWP in this simulation, while the smaller $ERF_{ari+aci}$ in the Northern Hemisphere Atlantic is due to a different reason. E63H23 uses the Köhler-theory based Abdul-Razzak and Ghan (2000) activation scheme, while E61H22 applies the empirical Lin and Leaitch (1997) activation scheme which depends only on the

number of aerosol particles and updraft velocities. A sensitivity simulation with the Lin and Leaitch (1997) activation scheme applied in E63H23 shows a negative $ERF_{ari+aci}$ between 15°N and 45°N in the Atlantic (Fig. S5). Therefore, the stronger $ERF_{ari+aci}$ in E61H22 can be partly explained by the different activation scheme. Another reason for the stronger $ERF_{ari+aci}$ in E61H22 over oceans between 15°N and 45°N is that different sea salt parameterizations are used in E61H22 and



E63H23. Tegen et al. (2018) show that the Long et al. (2011) sea salt parameterization (temperature dependent; Sofiev et al., 2011) used in E63H23 leads to higher aerosol number concentrations over ocean compared to the Guelle et al. (2001) sea salt parameterization used in E61H22, improving the agreement with measured sea salt surface concentrations and particle size distributions at different marine sites (see also the comparison of sea salt parameterizations in Zieger et al., 2017). The

higher natural background aerosol concentrations due to the higher sea salt aerosol number concentrations in E63H23 explain also why $ERF_{ari+aci}$ is less negative in E63H23 between 15°N and 45°N over oceans than in E61H22.

Most of the differences between the model versions discussed above are differences in SW $ERF_{ari+aci}$. There is one important difference in LW $ERF_{ari+aci}$ between the model versions. LW $ERF_{ari+aci}$ is more than twice as large as in E61H22 than in E55H20 and E63H23 (Table 3). The stronger LW $ERF_{ari+aci}$ in E61H22 occurs in Northern Hemisphere mid-latitudes and in

the tropics (Fig. 12). In Northern Hemisphere mid-latitudes also LW CRE is larger in E61H22 due to the ICNC bug (see Fig. 2; also ICNC itself is higher in E61H22, see Table 3). The strong LW $ERF_{ari+aci}$ in E61H22 is therefore likely an artefact which was removed in the latest model version.

Tegen et al. (2018) found an improved aerosol representation in biomass burning regions when GFAS biomass burning emissions, multiplied by a scaling factor of 3.4, as recommended by Kaiser et al. (2012), replace the default ACCMIP

biomass burning emissions. Therefore, we performed a E63H23 simulation with GFAS biomass burning emissions multiplied by 3.4 (E63H23-GFAS34). E63H23-GFAS34 has a weaker $ERF_{ari+aci}$ (-0.9 W m$^{-2}$) than E63H23 (-1.0 Wm$^{-2}$), because the pre-industrial aerosol burden is higher in E63H23-GFAS34 and $ERF_{ari+aci}$ is sensitive to pre-industrial aerosol concentrations (Carslaw et al., 2013). Also, the present day aerosol burdens in E63H23-GFAS34 agree better with the mean aerosol burden of the AeroCom models (Textor et al., 2006) than in E63H23 (see Table 3 and S1).

We would like to point out that our simulations include interactions between sulfate and mineral dust. On the one hand (anthropogenic and natural) gaseous sulfate may coat mineral dust particles which leads to a transfer of dust from insoluble modes to soluble modes in the models, which increases the wet deposition of dust (and leads to decreased present day mineral dust burdens, see Table S2), while on the other hand mineral dust particles provide surfaces where (anthropogenic and natural) gaseous sulfate may condensate, leading to a dampening of the nucleation of new particles. Similar interactions

between sulfate and mineral dust have been found by Bauer and Koch (2005) and Bauer et al. (2007) using the Goddard Institute for Space Studies (GISS) climate model, modelE (Schmidt et al., 2006; Hansen et al., 2005) or Fan et al. (2004). The forcing from these interactions between sulfate and mineral dust is included in our estimates for $ERF_{ari+aci}$ and $RF_{ari}$ (these interactions will make $ERF_{ari+aci}$ and $RF_{ari}$ less negative but they are difficult to quantify).

$RF_{ari}$ is shown in Fig. 12 for all-sky and clear-sky conditions for E63H23, E61H22 and E55H20 (since $RF_{ari}$ is computed by

double calls to the radiation scheme, many values in Fig. 12 are statistically significant). $RF_{ari}$ is strong in the east of North America, Europe, South Asia, East Asia and the tropical Atlantic and Indian Oceans. The differences in the strength of $RF_{ari}$ between E55H20 and E63H23 and E61H22 in these regions are predominantly due to different emission years (and a different emission dataset) used in E55H20, as described above for $ERF_{ari+aci}$. Differences in aerosol water uptake can explain the stronger $RF_{ari}$ over land in E63H23 than in E61H22. Absorbing aerosol above clouds leads to a positive $RF_{ari}$.





This can be seen in all three model versions in the all-sky RF$_{ari}$ fluxes west off Africa (in particular in the Southern Hemisphere) and to a lesser extent also west of South America. The significant positive RF$_{ari}$ in the Saharan region and the Arabian Peninsula in E55H20 is due to a coding error in E55H20 (the refractive index of POM was used for sulfate aerosol) which was fixed in later model versions. The small positive RF$_{ari}$ in the Saharan region, the Arabian Peninsula and Pakistan in E61H22 and E63H23 is due to a decrease in dust load, which is caused by interaction with sulfate aerosol as described above (also present in E55H20 but shadowed by the coding error). RF$_{ari}$ is weaker over ocean in E63H23 than in E61H22 and E55H20. One reason is that the dust burden is larger in E63H23 than in the other model versions and also the decrease in dust burden is larger in E63H23, leading to a positive RF$_{ari}$ which compensates the negative RF$_{ari}$ from the increase in anthropogenic aerosol. But there are also differences in aerosol water uptake (aerosol water increases less over oceans in E63H23 than in the other two model versions).

The equilibrium climate sensitivity (ECS) is strongest in E55H20 (3.5 K), weaker in E61H22 (2.8 K) and weakest in E63H23 (2.5 K) (Fig. 13). For E61H22 and E63H23 we computed the cloud feedback parameter using the cloud radiative kernel method of Zelinka et al. (2016) (Fig. 14; in E55H20 the COSP-ISCCP simulator is not implemented, therefore the cloud feedback parameter could not be computed). In addition, we computed ECS and cloud feedback for the E63H23-GFAS34 and E63H23-10CC simulations. E63H23-GFAS34 and E63H23 have very similar ECS and cloud feedback. In E63H23-10CC, on the other hand, ECS is stronger and the cloud feedback is more positive than in E63H23 (leading to more warming in agreement with the stronger ECS). The optical depth feedback of low clouds is more positive between 40°N and 40°S in E63H23-10CC. The optical depth feedback of non-low clouds is less negative in the tropics and in mid-latitudes. This could be an indication of a weaker cloud phase feedback. As there are fewer but larger cloud droplets in E63H23-10CC than in E63H23, the cloud droplets have a shorter lifetime and this decreases differences between ice clouds and liquid water clouds. A similar less negative cloud optical depth feedback of non-low clouds (weaker cloud phase feedback) occurs in E61H22 (CDNC are higher and the representation of supercooled liquid in mixed-phase clouds is improved in E63H23 compared to E61H22, see Fig. S6). Furthermore, in E61H22 also the cloud amount feedback of low clouds is more positive than in E63H23. This is because in E63H23 in regions of low cloud cover where shallow convective clouds are simulated, the cloud amount feedback is negative whereas in E61H22 it is positive. This seems to be related to the stronger entrainment rate for shallow convection in E63H23 (Mauritsen et al., 2012). When more of the water vapor remains in the boundary layer as in E63H23, the increased water vapor in the warmer climate can lead to increased cloud cover. The overall more positive cloud feedback in E61H22 than in E63H23 agrees with the stronger ECS in E61H22.

The largest differences between E61H22 and E63H23 in terms of ERF$_{ari+aci}$ are therefore due to a more realistic simulation of cloud water, the removal of a bug in ICNC, the new activation scheme and the new sea salt emission parameterization in E63H23, whereas for ECS they are due to a more realistic simulation of cloud water, and to model tuning.



## 3.6 Impact of changes and improvements in E63H23

The liquid phase of clouds is better represented in E63H23 than in the previous model versions because the low-bias in cloud cover in the subtropics is reduced and the zonal distribution of LWP agrees with observations. This leads also to a better agreement of the SW CRE with observations in E63H23. Important reasons for these improvements are the change in the

fractional cloud cover scheme for marine stratocumulus clouds and the removal of an inconsistency which had led to either 0 or 1 cloud cover in ECHAM6.3 and subsequent changes in model tuning. Furthermore E63H23 uses the Abdul-Razzak and Ghan (2000) activation scheme and the Long et al. (2011) sea salt emission parameterization (temperature dependent; Sofiev et al., 2011) which leads to higher CDNC concentrations where LWP is large. The Abdul-Razzak and Ghan (2000) activation scheme is more physically realistic as the empirical Lin and Leaitch (1997) activation scheme used in the previous

model versions as it is Köhler-theory based and therefore takes into account the size of the aerosol particles and their chemical composition. Although the Abdul-Razzak and Ghan (2000) activation scheme has limitations under certain conditions (the assumption that the aerosol particles are in equilibrium with its environment is not valid in all conditions; Phinney et al., 2003) and does not account for pre-existing cloud droplets during cloud droplet activation (Barahona et al., 2010), it certainly helps to improve the representation of cloud droplets in E63H23. The performance of the new Long et al.

(2011) (temperature dependent; Sofiev et al., 2011) and the old Guelle et al. (2001) sea salt parameterizations in E63H23 was analyzed by Tegen et al. (2018). The new temperature dependence leads to increased sea salt emissions where the sea surface temperature is warmer than 20°C and a decrease at colder temperatures. The new sea salt parameterization performs better, particular with respect to number concentrations, than the previous sea salt parameterization compared to measurements on research cruises and research stations. This is another indication that CDNC concentrations are more

realistic in E63H23 than in the previous model versions. The higher CDNC concentrations in E63H23 allowed us to increase the tuning parameter for autoconversion of cloud droplets to rain. Together these changes led to a better representation of the liquid phase of clouds in E63H23 and to a reduction of the SW component of ERF$_{ari+aci}$ in E63H23 compared to E61H22 (because CCN concentrations from natural background aerosol are higher in E63H23).

Also the ice phase of clouds has improved in E63H23 compared to previous model versions. The low-bias in IWP is reduced

in E63H23 and the global mean vertical IWC is within the observational range (Fig. 7). This is because the Seifert and Beheng (2006) sticking efficiency used in E63H23 leads to a less efficient removal of ice crystals by snow. A subsequent reduction in the tuning parameter for stratiform snow formation by aggregation further increases IWP in E63H23. Only few laboratory studies for sticking efficiency have been conducted and even fewer theories for sticking efficiency were developed (Phillips et al, 2015). We find that the simple formulation of Seifert and Beheng (2006) for sticking efficiency for

accretion of ice crystals by snow improves the simulation of cloud ice in E63H23. Furthermore, the altitude of the global mean maximum IWC agrees well with observations in E63H23 whereas in E61H22 and E55H20 it is at higher altitudes than observed. This can be explained by changes in ICNC described in section 2.1.5 such as the use of a consistent ice crystal shape (hexagonal plates), removal of an ICNC bug or the changed treatment of detrained ice crystals. The subsequent



changes in precipitation formation and ice crystal sedimentation can then lead to a different vertical distribution of cloud ice. In E61H22 the global ICNC burden is considerably higher than in the two other model versions because of an inconsistency between cloud droplet activation, condensation, vertical transport of CDNC and homogeneous freezing of cloud droplets in cirrus clouds, which led to homogeneous freezing of aerosol particles even when the water vapor pressure was too low for

homogeneous nucleation. The higher ICNC are also responsible for the LW component of $ERF_{ari+aci}$ in E61H22 being more than twice as large as in the other two model versions. The good agreement of the global mean vertical distribution of IWC and LW CRE with observations in E63H23 is an indication that ICNC and the size of ice crystals are closer to reality in E63H23 than in the previous model versions. In a future version of the model we want to include also the competition for water vapor between homogeneous freezing of solution droplets, heterogeneous freezing of ice nucleating particles and pre-

existing ice crystals in cirrus cloud formation as has been done by Kuebbeler et al. (2014) or Gasparini et al. (2018), which should further improve the simulation of ICNC.

While the global mean values of $RF_{ari}$ are quite similar between E63H23 and the previous model versions, there are regional differences. These are caused by the removal of inconsistencies in the model code and for E63H23 also by the new emission parameterization for mineral dust which uses new satellite-based data for dust sources, which increases the confidence in the

simulation of $RF_{ari}$ in E63H23.

The weaker ECS in E63H23 compared to E61H22 can be linked to changes in cloud feedbacks. There are indications for a stronger cloud phase feedback in non-low clouds due to increased CDNC and changes in cloud water in E63H23. A stronger (cooling) cloud phase feedback will lead to less warming in the future. Similarly the less positive cloud amount feedback of low clouds (related to model tuning in ECHAM6.3) in E63H23 contributes to the weaker ECS in E63H23 compared to

E61H22.

The changes and improvements in E63H23 therefore have not only improved the representation of clouds in E63H23 compared to previous model versions, they also have an impact on $ERF_{ari+aci}$ and ECS, decreasing both.

## 4 Summary and conclusions

Clouds in the current (E623H23) and previous (E55H20 and E61H22) model versions of the ECHAM-HAM global aerosol-

climate model were evaluated using a suite of global observational datasets for clouds and precipitation. Improvements in E63H23 compared to previous model versions for cloud water include:

- a more physically based activation scheme,

- changes in the treatment of CDNC detrained from convective clouds,

- an increase in low clouds,

- which together lead to a more realistic LWP globally.

For cloud ice the improvements include:

- a different sticking efficiency for accretion of ice crystals by snow,



- consistent ice crystal shapes throughout the model,
- changes in mixed phase freezing,
- the removal of an inconsistency in ICNC in cirrus clouds,
- which together lead to a more realistic IWP globally.

The sum of the changes leads to improved cloud radiative effects. Although the representation of shallow convective clouds has improved in E63H23, stratocumulus clouds are still underrepresented. The comparison of the different model versions showed that the misrepresentation of certain cloud types can lead to compensating biases in other clouds via model tuning. Therefore, if the bias in stratocumulus clouds in E63H23 can be reduced, this could also improve the representation of other cloud types. Reasons for the bias in stratocumulus clouds identified by Neubauer et al. (2014) in E61H22 were e.g. too

strong turbulent mixing at cloud top, the shallow convection scheme triggering too often or a lack of vertical resolution. Future work will focus on addressing these difficult issues.

Deep convective clouds in the Atlantic and Pacific oceans form too close to the continents, which leads to biases in the geographical distribution of precipitation. While the geographical (except for deep convective clouds) and vertical distribution of cloud ice agree well with observations in E63H23, IWP remains biased low. The combination of observations

of IWP, LW CRE and the vertical distribution of cloud ice indicate that ICNC may be overestimated in ECHAM-HAM. Previous studies with ECHAM-HAM showed that the bias in ICNC and IWP can be reduced when heterogeneous freezing of ice nucleating particles and/or water vapor deposition on pre-existing ice crystals are accounted for in cirrus clouds.

Estimates of $ERF_{ari+aci}$ and ECS of E55H20, E61H22 and E63H23 were compared since the representation of clouds is important for both $ERF_{ari+aci}$ and ECS. The largest differences between E61H22 and E63H23 in terms of SW $ERF_{ari+aci}$ are

due to:

- the new activation scheme in E63H23,
- the new sea salt emission parameterization,
- the more realistic simulation of cloud water

which lead to a weaker SW $ERF_{ari+aci}$ in E63H23. In terms of LW $ERF_{ari+aci}$ the difference is due to:

- the removal of an inconsistency in ICNC in cirrus clouds leading to a weaker LW $ERF_{ari+aci}$ in E63H23.

Since there are reductions in both SW and LW $ERF_{ari+aci}$ the net $ERF_{ari+aci}$ is only slightly weaker in E63H23 (-1.0 W m$^{-2}$). A sensitivity simulation where CDNCmin was lowered to 10 cm$^{-3}$ leads to a stronger $ERF_{ari+aci}$ everywhere (-1.7 W m$^{-2}$) showing that the necessary usage of CDNCmin (Lohmann and Neubauer, 2018) has a strong impact on $ERF_{ari+aci}$. Another sensitivity simulation with increased biomass burning emissions (E63H23-GFAS34) indicates that $ERF_{ari+aci}$ in E63H23

would be weaker (-0.9 W m$^{-2}$) when the representation of biomass burning aerosol could be improved.

ECS is weaker in E63H23 (2.5 K) than in E61H22 (2.8 K) (and E55H20; 3.5 K). The decrease compared to E61H22 is due to:


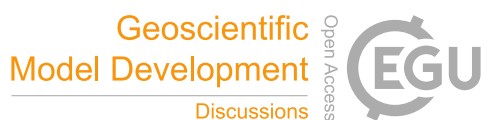

- changes in the entrainment rate for shallow convection (which leads to a less positive feedback of cloud amount of low clouds in some regions),
- a stronger cloud phase feedback.

Although the differences in both $ERF_{ari+aci}$ and ECS between E63H23 and E61H22 can be explained by changes in the
representation of clouds, not the same changes in the clouds that affect $ERF_{ari+aci}$ also affect ECS and vice versa. Therefore, many aspects of clouds in GCMs will need to be improved to increase the confidence in computations of $ERF_{ari+aci}$ and ECS.

*Code availability.* The ECHAM-HAMMOZ model is made freely available to the scientific community under the HAMMOZ Software Licence Agreement, which defines the conditions under which the model can be used. More
information can be found at the HAMMOZ Website (https://redmine.hammoz.ethz.ch/projects/hammoz).
Scripts can be found at http://doi.org/10.5281/zenodo.2553892 (Neubauer et al., 2019a).

*Data availability.* Data can be found at http://doi.org/10.5281/zenodo.2541937 (Neubauer et al., 2019b). ESA Cloud CCI data can be downloaded from: http://www.esa-cloud-cci.org/?q=data_download. MODIS products are available for
download from Level 1 and Atmosphere Archive and Distribution System (LAADS) https://ladsweb.modaps.eosdis.nasa.gov/search/. ISCCP histogram data and CALIPSO-GOCCP product can be obtained from http://climserv.ipsl.polytechnique.fr/cfmip-obs/. Cloud top CDNC can be downloaded from https://doi.org/10.15695/vudata.ees.1. MAC-LWP data is available at the Goddard Earth Sciences Data and Information Services Center (GES DISC, current hosting: http://disc.sci.gsfc.nasa.gov). CERES satellite data can be obtained from the
NASA Langley Research Center Atmospheric Science Data Center https://ceres.larc.nasa.gov/order_data.php. The IWP satellite data from Li et al. (2012) was obtained from the authors. GPCP data can be downloaded from https://www.esrl.noaa.gov/psd/data/gridded/data.gpcp.html.

*Author contribution.* DN designed the evaluation methodology with comments from co-authors. DN performed the
experiments and the analysis of the data. CS and SF provided support needed to run the ECHAM-HAM model versions. All co-authors are involved in the development of the ECHAM-HAMMOZ model. DN wrote the paper with comments from co-authors.

*Competing interests.* The authors declare no competing interests are present.

*Acknowledgements.* The ECHAM-HAMMOZ model is developed by a consortium composed of ETH Zurich, the Max Planck Institut für Meteorologie, Forschungszentrum Jülich, the University of Oxford, the Finnish Meteorological Institute and the Leibniz Institute for Tropospheric Research, and managed by the Center for Climate Systems Modeling (C2SM) at ETH Zurich. This study has received partly funding from the Center for Climate System Modelling (C2SM) at ETH Zurich,





from the Swiss National Science Foundation (project number 200021_160177), Academy of Finland projects no. 308292 and 307331 and from the European Union's Seventh Framework Programme (FP7/2007-2013) project BACCHUS under grant agreement no. 603445. P.S. acknowledges funding from the European Research Council project RECAP under the European Union's Horizon 2020 research and innovation programme with grant agreement 724602. The computing time for

this work was supported by a grant from the Swiss National Supercomputing Centre (CSCS) under project ID s652 and from ETH Zurich. The CERES satellite data were obtained from the NASA Langley Research Center Atmospheric Science Data Center. We would like to thank Sebastian Rast for useful comments and suggestions.

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



Table 1. Setup of the simulations for E55H20, E61H22 and E63H23.

| Simulation | Configuration | E55H20 | E61H22 | E63H23 |
|---|---|---|---|---|
| All | Resolution | T63L31 | T63L31 | T63L47 |
| | Sea salt emissions | Guelle et al. (2001) | Guelle et al. (2001) | Long et al. (2011) / Sofiev et al. (2011) |
| | Oxidants | MOZART (present day, ~2000) | MACC (2003-2010 mean) | MACC (2003-2010 mean) |
| PD/PD$_{aer}$/PI$_{aer}$ | Greenhouse gases | RCP8.5 (year 2008) | | |
| PD | Simulation period | 2000-2009 | 2000-2009 | 2003-2012 |
| | Aerosol emissions | AEROCOM I (year 2000) | ACCMIP MACCity (historic and RCP8.5) | ACCMIP MACCity (historic and RCP8.5) |
| | SST and SIC | AMIP 2000-2015 mean | | |
| PD$_{aer}$ | | Identical to PD except for simulation period | | |
| | Simulation period | 2000-2019 | | |
| PI$_{aer}$ | | Identical to PD$_{aer}$ except for aerosol emissions | | |
| | Aerosol emissions | AEROCOM I (year 1750) | ACCMIP MACCity (year 1850) | ACCMIP MACCity (year 1850) |
| 1xCO$_2$/2xCO$_2$ | Simulation duration | 50 years (25 years spin-up, 25 years for analysis) | | |
| | Aerosol emissions | Same as in PI$_{aer}$ | | |
| | SST and SIC | Mixed-layer ocean (50 m deep) | | |
| | Heat flux corrections | Computed from extended PD$_{aer}$ (2000-2024) | | |
| 1xCO$_2$ | Greenhouse gases | Year 1850 CO$_2$: 284.7 ppm | | |
| 2xCO$_2$ | Greenhouse gases | Year 1850 CO$_2$: 569.5 ppm | | |



Table 2. Parameter settings for E55H20, E61H22 and E63H23. The parameters used to tune the ECHAM-HAM versions are stratiform rain formation rate by autoconversion ($\gamma_r$), stratiform snow formation rate by aggregation ($\gamma_s$), critical relative humidity at the surface which is used in the cloud cover scheme ($\gamma_c$), entrainment rate for shallow convection ($\epsilon_s$), entrainment rate for deep convection ($\epsilon_d$), convective conversion rate from cloud water to rain ($\gamma_{cr}$), inhomogeneity factor for ice clouds ($\gamma_i$) and the minimum cloud droplet number concentration (CDNCmin).

| Parameter | E55H20 | E61H22 | E63H23 |
|---|---|---|---|
| $\gamma_r$ | 3 | 4 | 10.6 |
| $\gamma_s$ | 1200 | 1200 | 900 |
| $\gamma_c$ | 0.9 | 0.9 | 0.975 |
| $\epsilon_s$ (m$^{-1}$) | 0.0003 | 0.0008 | 0.003 |
| $\epsilon_d$ (m$^{-1}$) | 0.0001 | 0.00035 | 0.0002 |
| $\gamma_{cr}$ (s$^{-1}$) | 0.0001 | 0.0009 | 0.0009 |
| $\gamma_i$ | 0.85 | 0.7 | 0.7 |
| CDNCmin (cm$^{-3}$) | 20 | 40 | 40 |



Table 3. Global mean values of the PD simulations. Radiative fluxes are at the top of atmosphere. Values from observations (OBS) and multi-model means (MMM) for aerosol burdens are shown next to those of the three model versions. $ERF_{ari+aci}$ and ECS are from the $PD_{aer}/PI_{aer}$ and 1xCO2/2xCO2 simulations respectively.

| Variable | OBS/MMM | E55H20 (2000-2009) | E61H22 (2000-2009) | E63H23 (2003-2012) |
|---|---|---|---|---|
| SW (W m$^{-2}$) | 241 (238 to 244)[a] | 232 | 236 | 238 |
| LW (W m$^{-2}$) | -240 (-237 to -241)[a] | -232 | -236 | -238 |
| Net(W m$^{-2}$) | 0.7±0.1[b] | -0.1 | 0.4 | 0.4 |
| SW CRE (W m$^{-2}$) | -46 (-44 to 53.3)[c] | -53 | -52 | -50 |
| LW CRE (W m$^{-2}$) | 28 (22 to 30.5)[c] | 28 | 27 | 24 |
| Net CRE (W m$^{-2}$) | -18 (-17.1 to 22.8)[c] | -25 | -25 | -26 |
| CC (%) | 68±5[d] | 64 | 64 | 69 |
| LWP (ocean) (g m$^{-2}$) | 42.9 to 89.4[e] | 85 | 94 | 71 |
| LWP-LP (ocean) (g m$^{-2}$) | 73.5±5.5[f] | 104 | 96 | 76 |
| IWP (g m$^{-2}$) | 25±7[g] | 8 | 10 | 15 |
| Cloud-top CDNC (ocean; 60°N-60°S) (cm$^{-3}$) | 72±37[h] | 80 | 76 | 78 |
| CDNC$_{burden}$ (10$^{10}$ m$^{-2}$) | - | 3.1 | 3.2 | 3.1 |
| ICNC$_{burden}$ (10$^{12}$ m$^{-2}$) | - | 8.9 | 17.9 | 8.0 |
| P (mm d$^{-1}$) | 2.7±0.2[i] | 3.0 | 3.0 | 3.0 |
| Sulfate burden (Tg) | 2.0(±25%)[j] | 2.6 | 1.9 | 2.2 |
| Black carbon burden (Tg) | 0.2(±42%)[j] | 0.13 | 0.15 | 0.14 |
| Particulate organic matter burden (Tg) | 1.7(±27%)[j] | 1.1 | 1.1 | 1.0 |
| Sea salt burden (Tg) | 7.5(±54%)[j] | 12.6 | 10.8 | 4.1 |
| Mineral dust burden (Tg) | 19.2(±40%)[j] | 8.0 | 10.9 | 18.2 |
| Aerosol water burden (Tg) | 27.7(±46%)[j] | 48.4 | 48.9 | 23.0 |
| RF$_{ari}$ (all-sky) (W m$^{-2}$) | -0.27±0.15[k] | -0.04 | -0.06 | 0.00 |
| RF$_{ari}$ (clear-sky) (W m$^{-2}$) | -0.67±0.18[k] | -0.41 | -0.30 | -0.27 |
| ERF$_{ari+aci}$ (W m$^{-2}$) | -0.9 (-1.9 to -0.1)[l] | -1.1 | -1.2 | -1.0 |
| SW ERF$_{ari+aci}$ (W m$^{-2}$) | - | -1.3 | -2.0 | -1.3 |
| LW ERF$_{ari+aci}$ (W m$^{-2}$) | - | 0.2 | 0.8 | 0.3 |
| ECS (K) | 1.5 to 4.5[m] | 3.5 | 2.8 | 2.5 |

5  [a] Central values from Loeb et al. (2018), range from Stevens and Schwartz (2012). [b] Loeb et al. (2018) and Johnson et al. (2016). [c] Central values from Loeb et al. (2018), the range takes into account values from Loeb et al. (2009) and Matus and



L'Ecuyer (2017). [d] Stubenrauch et al. (2013). [e] Platnick et al. (2015, 2017), ATSR2-AATSR v2.0 (Stengel et al., 2017a; Poulsen et al., 2017), Elsaesser et al. (2017). [f] Elsaesser et al. (2017). [g] Li et al. (2012). [h] Bennartz and Rausch (2017). [i] Central value from Adler et al. (2018), uncertainty from Adler et al. (2012). [j] Taken from Table 10 of Textor et al. (2006). [k] Taken from Table 3 of Myhre et al. (2013). [l] Boucher et al. (2013). [m] Collins et al. (2013), Knutti et al. (2017).





**Figure 1: Comparison of zonal annual mean values of E55H20, E61H22 and E63H23 to observations, (a) SW CRE, (b) LWP-low precipitation over oceans, (c) LW CRE, (d) IWP, (e) total cloud cover, (f) Cloud top CDNC of clouds between 268 and 300 K over oceans. Observations of IWP are from Li et al. (2012), LWP-low precipitation over oceans from Elsaesser et al. (2017), Cloud top CDNC over oceans from Bennartz and Rausch (2017). The solid SW and LW CRE lines are from CERES (Loeb et al., 2018), the dashed ones from ERBE (Barkstrom, 1984) and the dotted one for LW CRE is from TOVS satellite data (Susskind et al., 1997). Total cloud cover from CALIPSO GOCCP (solid line; Chepfer et al., 2010), AVHRR-PM (dashed line; Stengel et al., 2017b) and MODIS collection 6.1 (dotted line; Platnick et al., 2015, 2017).**



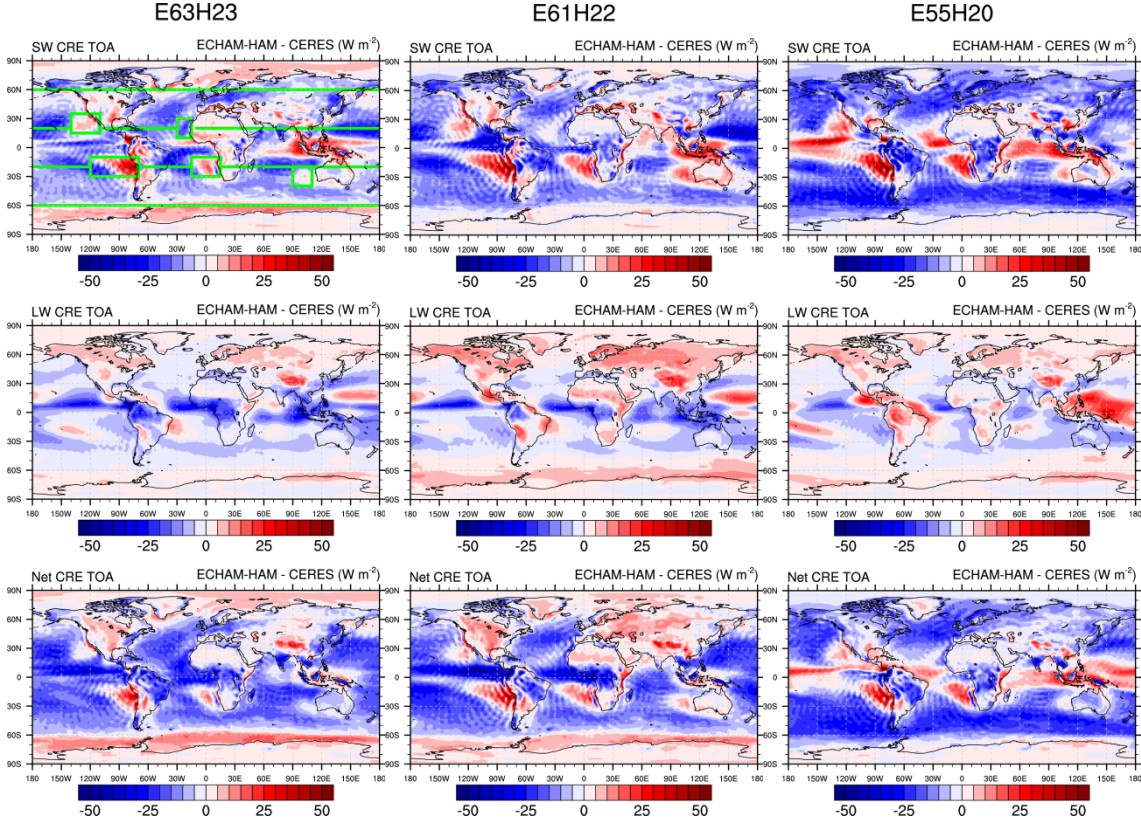

**Figure 2: Comparison of annual mean SW, LW and net CRE of E55H20, E61H22 and E63H23 to CERES 4.0 (Loeb et al., 2018) observations. CERES data is for 2005-2015, model data is from the PD simulations. In the top left panel the regions used for cloud top pressure vs. cloud optical depth histograms are shown by green lines.**





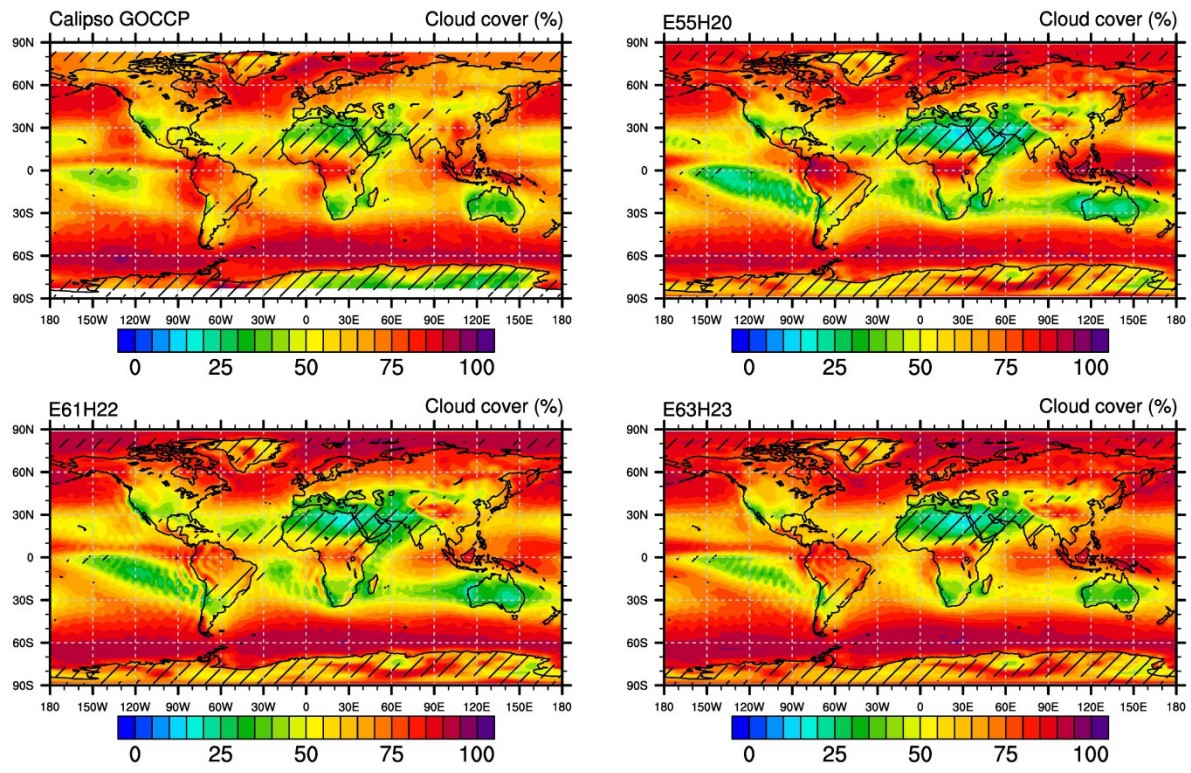

**Figure 3: Comparison of annual mean cloud cover of E55H20, E61H22 and E63H23 to CALIPSO GOCCP observations. Areas where the cloud cover of CALIPSO GOCCP, MODIS collection 6.1 and AVHRR-PM differ by more than five percent points are hatched. CALIPSO GOCCP data is for 2006-2010, model data is from the PD simulations.**





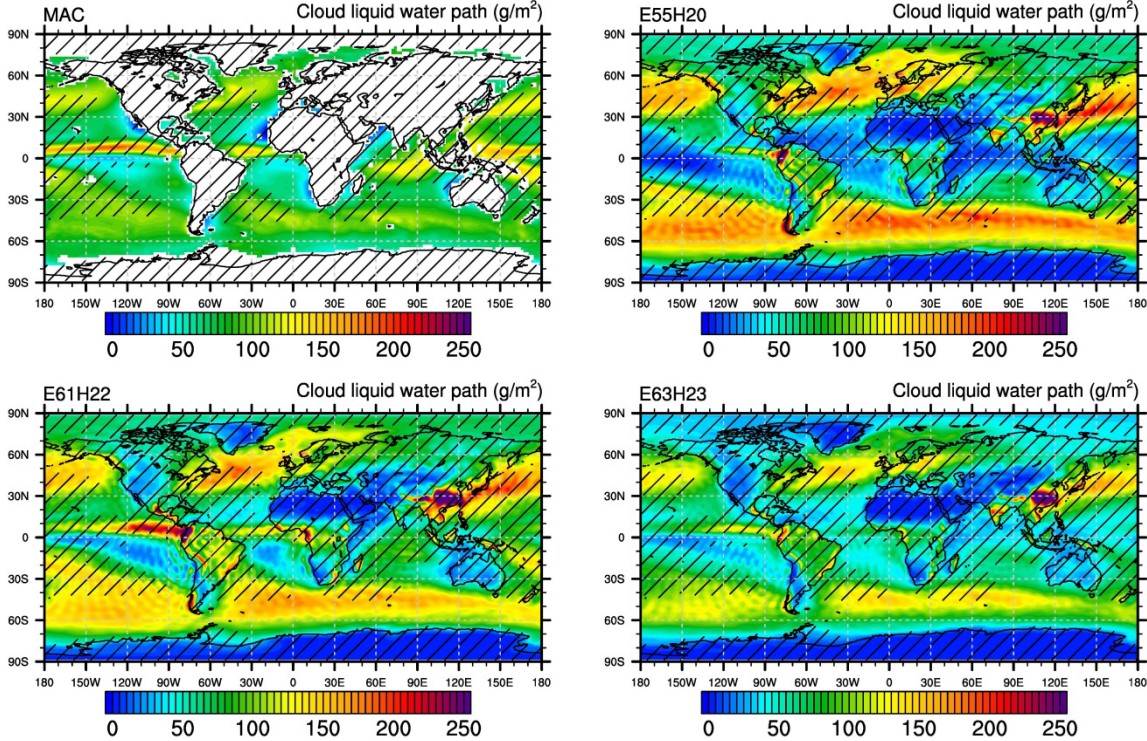

**Figure 4: Comparison of annual mean LWP of E55H20, E61H22 and E63H23 to MAC-LWP observations. Areas where precipitation could influence the LWP retrieval (LWP/(LWP+rain water path)≤0.8) are hatched. MAC data is for 2003-2012, model data is from the PD simulations.**





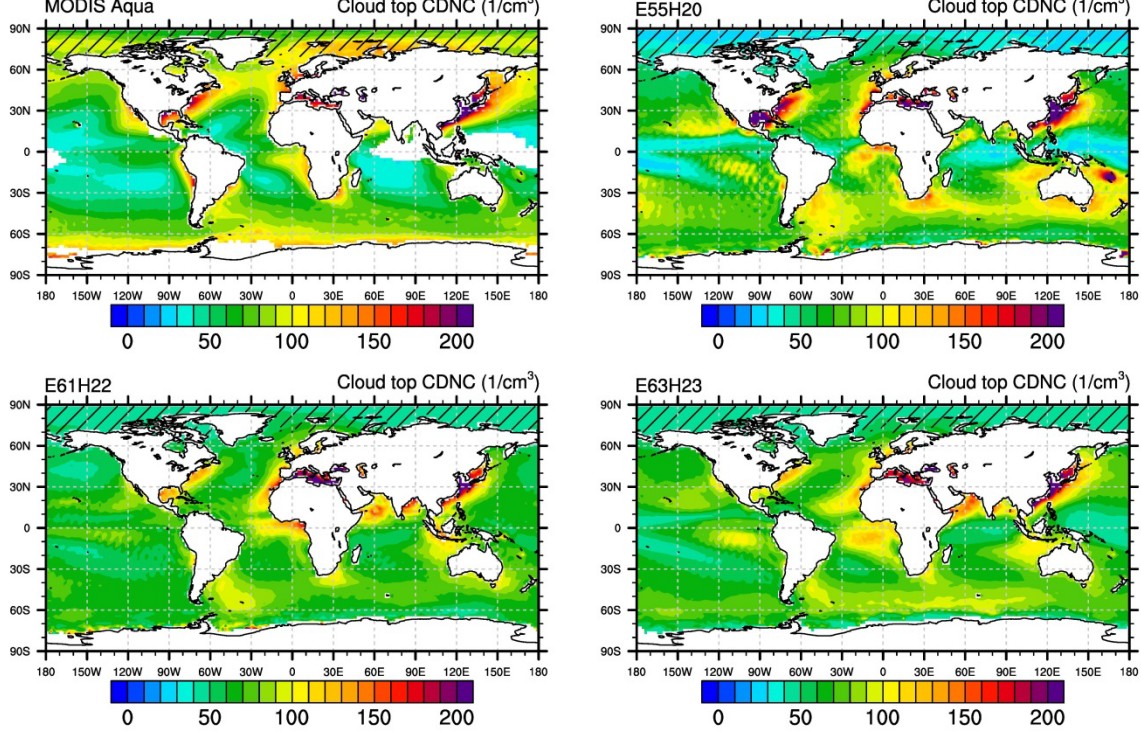

**Figure 5: Comparison of annual mean Cloud top CDNC of E55H20, E61H22 and E63H23 to MODIS observations from Bennartz and Rausch (2017). Areas where the relative uncertainty in the observations is larger than 75% are hatched. The MODIS data is for 2003-2015, model data is from the PD simulations.**



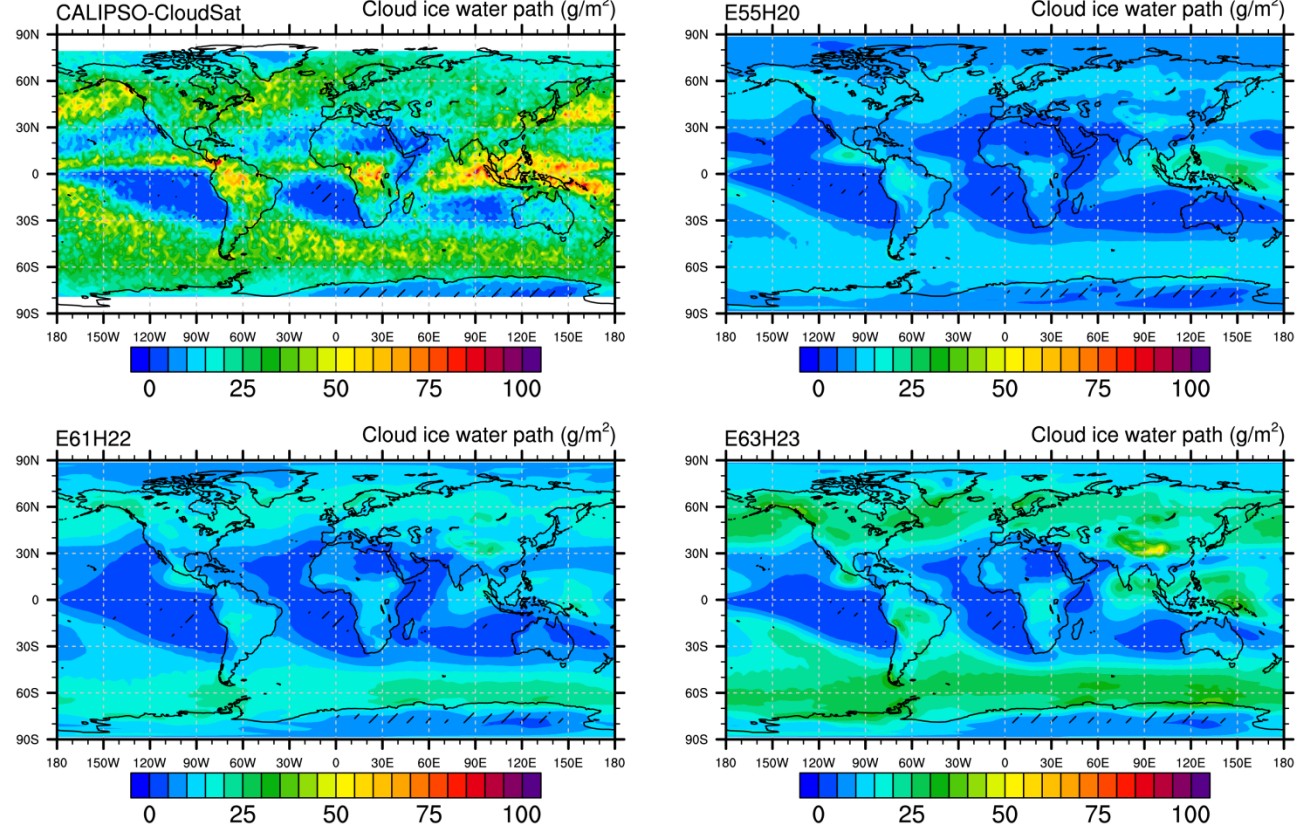

**Figure 6: Comparison of annual mean IWP of E55H20, E61H22 and E63H23 to CALIPSO/CloudSat observations from Li et al. (2012). Areas where the relative standard deviation of the different datasets compiled in Li et al. (2012) is larger than 75% are hatched. The CALIPSO/CloudSat data covers the years 2006-2010, model data is from the PD simulations.**



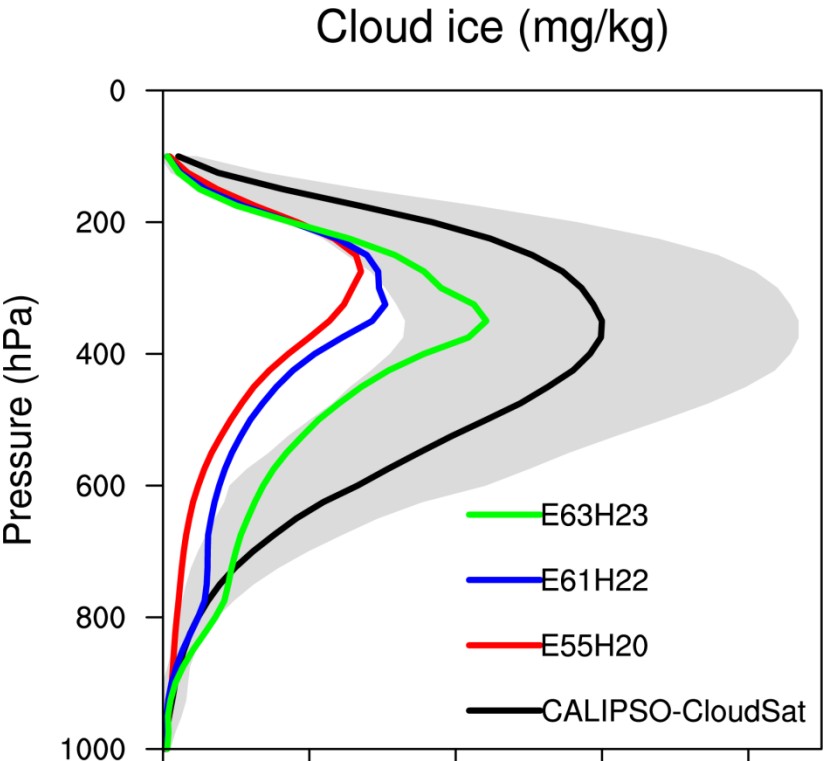

**Figure 7: Comparison of global annual mean IWC as a function of pressure of E55H20, E61H22 and E63H23 to CALIPSO/CloudSat observations from Li et al. (2012). Gray shading indicates the uncertainty in the CALIPSO/CloudSat observations. The CALIPSO/CloudSat data covers the years 2006-2010, model data is from the PD simulations.**





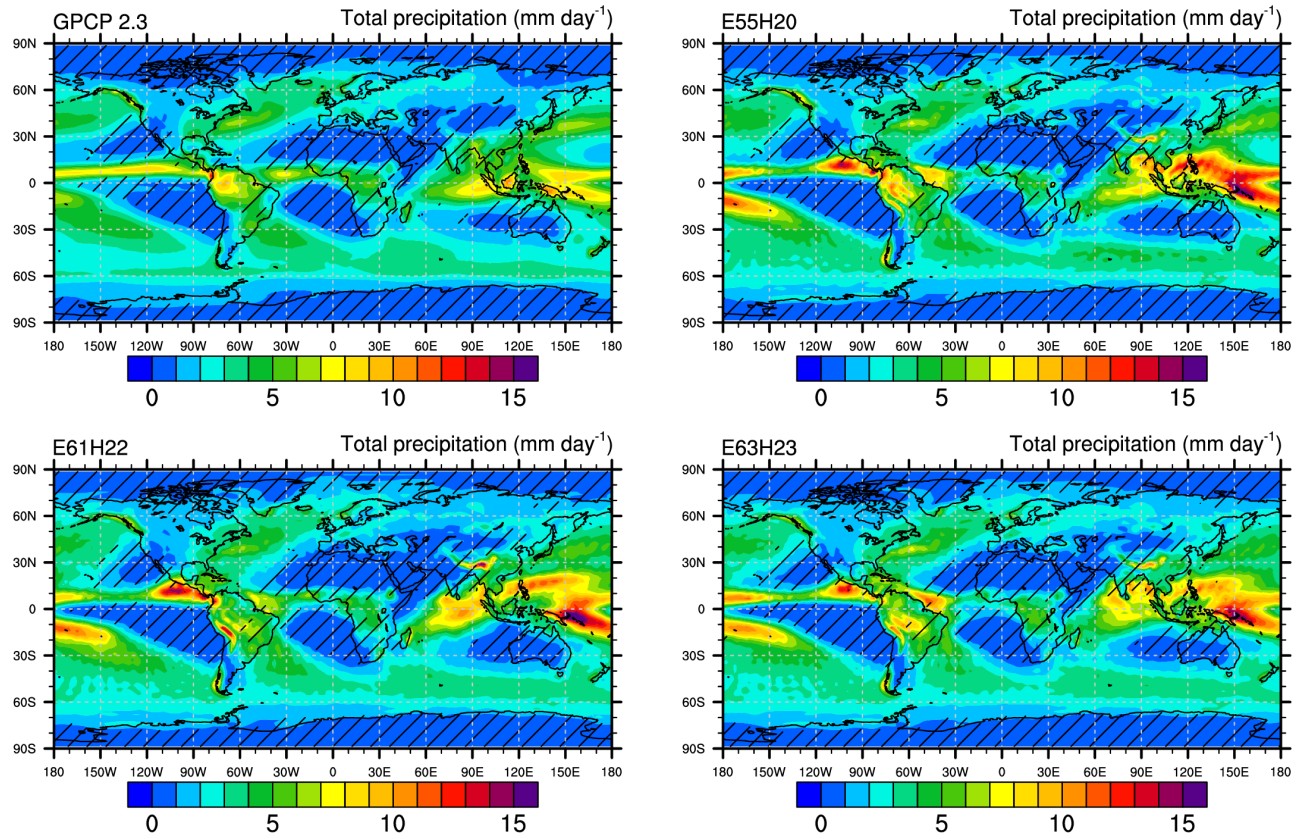

**Figure 8: Comparison of annual mean precipitation (stratiform + convective) of E55H20, E61H22 and E63H23 to GPCP2.3 observations. Areas where the relative uncertainty of the GPCP2.3 data is larger than 75% are hatched. The GPCP2.3 data is for 1979-2017, model data is from the PD simulations.**





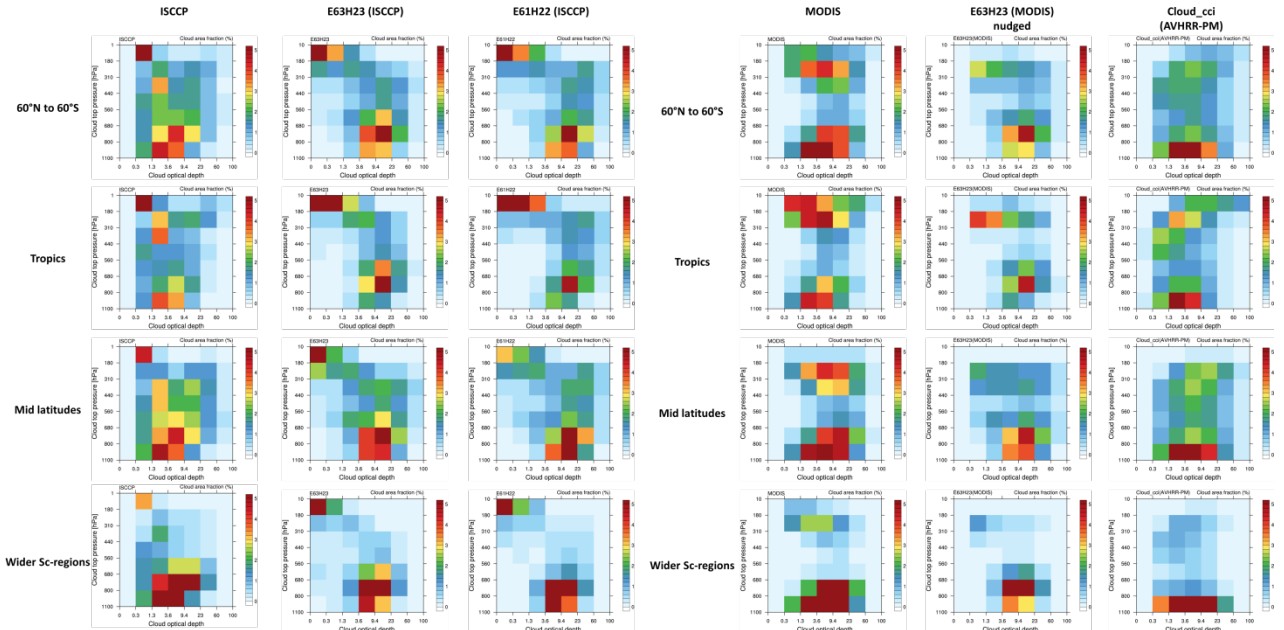

**Figure 9: Histograms of cloud top pressure vs. cloud optical depth of E61H22 and E63H23 as compared to ISCCP, MODIS and AVHRR-PM observations for different regions. The definition of the four regions shown is described in the text and the regions are shown in Fig. 2. The ISCCP data is for 2000-2008, MODIS data is for 2003-2012, AVHRR-PM data is for 2003-2012 and the model data is from the PD simulations.**





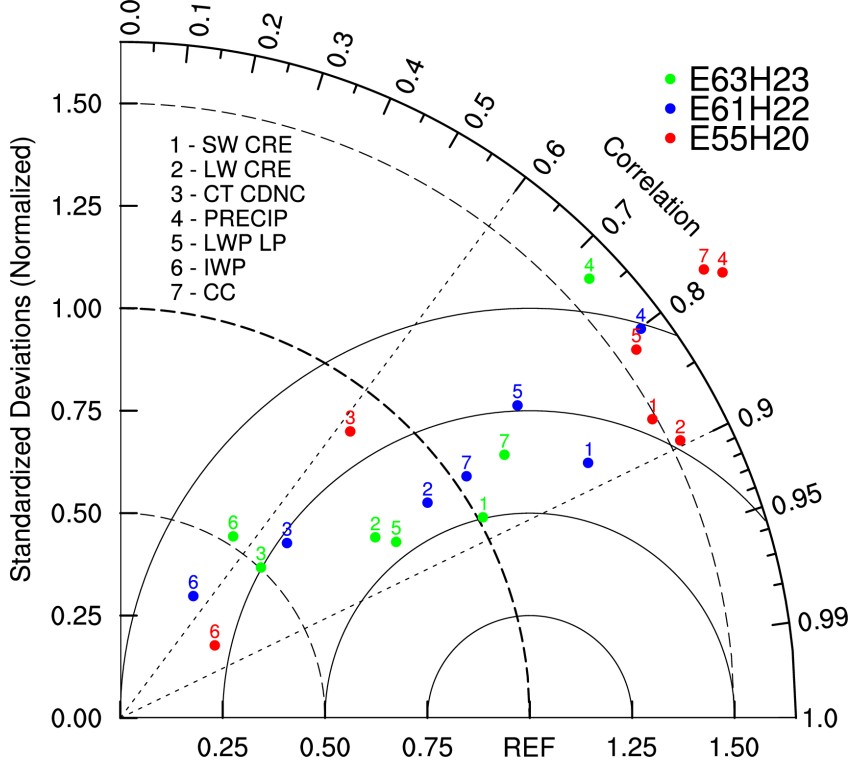

**Figure 10: Taylor diagram for comparison of SW and LW CRE, cloud cover, LWP-low precipitation, Cloud top CDNC, IWP and precipitation of E55H20, E61H22 and E63H23 to observations as REF. The standardized deviations of LWP-low precipitation are scaled by a factor of 1/3 to fit on the diagram. Only areas that are not hatched in Figs. 3, 6 and 8 were used to create the Taylor diagram. Observations are the same as in Figs. 3, 6 and 8. The correlation coefficient is shown as an angle and quantifies the similarity in pattern between modelled and observed annual mean fields. The standard deviation of the modelled fields (normalized by the standard deviation of the observed fields) is shown as the radial distance from the origin. The RMS error is shown as solid black circles and is the distance from the point marked by REF (the closer a model is to REF the better its skill to reproduce the observations).**

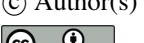



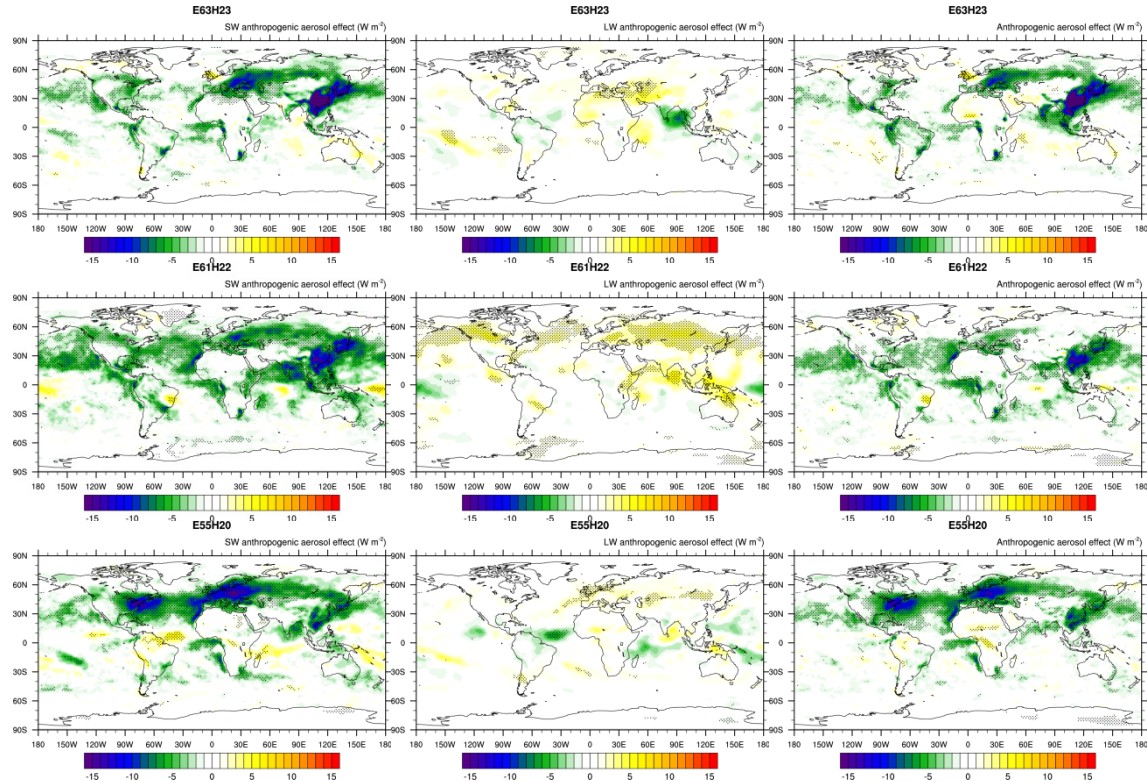

**Figure 11: Global maps of SW, LW and net ERF$_{ari+aci}$ of E55H20, E61H22 and E63H23 from 20 year free simulations with present day minus pre-industrial aerosol emissions (PD$_{aer}$-PI$_{aer}$). Hatching indicates statistically significant differences at the 95% significance level. The false discovery rate is controlled following Wilks (2016).**



**Figure 12: Global maps of all-sky and clear-sky net RF$_{ari}$ of E55H20, E61H22 and E63H23 from 20 year free simulations with present day minus pre-industrial aerosol emissions (PD$_{aer}$-PI$_{aer}$). Hatching indicates statistically significant differences at the 99% significance level. The false discovery rate is controlled following Wilks (2016).**





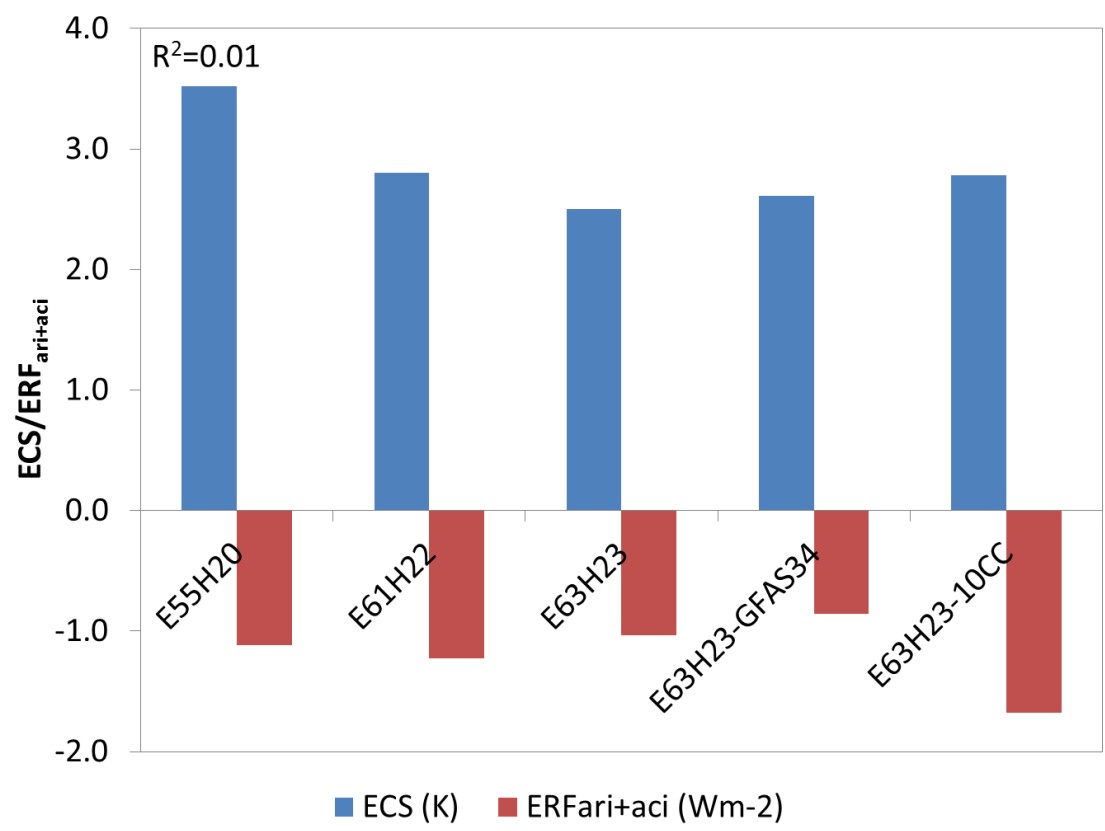

**Figure 13: Global mean ERF$_{ari+aci}$ and ECS of E55H20, E61H22, E63H23, E63H23-GFAS34 and E63H23-10CC. The coefficient of determination between ERF$_{ari+aci}$ and ECS is also displayed.**





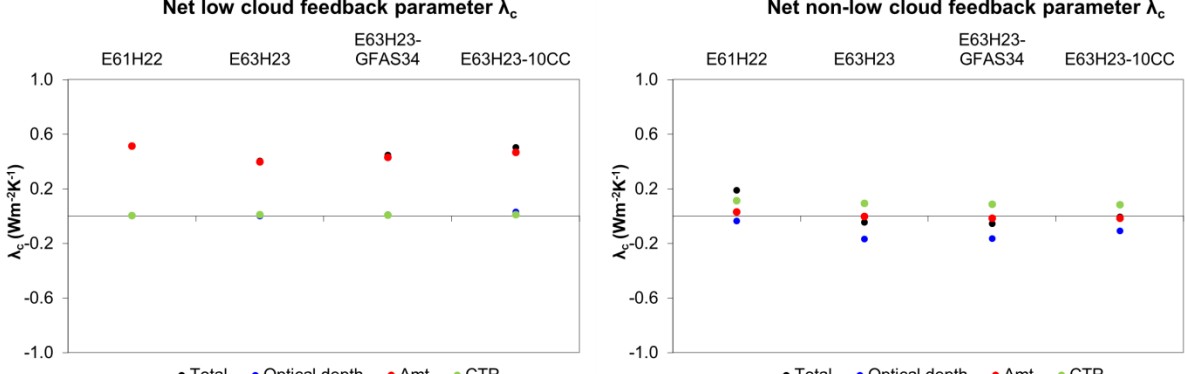

**Figure 14: Components of the net global mean cloud feedback parameter of E61H22 and E63H23 for low (cloud top pressure (CTP)>680hPa) and non-low (CTP<680 hPa) clouds.**