# Peer review of "The global aerosol-climate model ECHAM6.3-HAM2.3 – Part 2: Cloud evaluation, aerosol radiative forcing and climate sensitivity"

_Geoscientific Model Development, 2018_

## Referee Comment (RC1) · Anonymous Referee #1 · 25 Apr 2019

Neubauer et al. evaluate important aspects of a new version of a three-dimensional global aerosol-atmosphere model. It is well understood that cloud condensation nuclei concentrations vary significantly between strongly and less strongly polluted conditions and that radiative forcing by cloud-aerosol interactions depends non-linearly on cloud condensation nuclei concentration. At a time when box-model studies of cloud-aerosol interactions inform energy budget box-model studies, I find this study by Neubauer et al. a truly laudable effort. My only major concern with this study is that even in the latest model version, the minimum CDNC number is still artificially set to 40 per cubic centimetre. The authors discuss this point in some detail. They show that ER-Fari+aci and ECS depend on this threshold, but they do not mention this result in the

abstract. Although one can certainly argue about this point, I find the reasons for applying this particular threshold unconvincing. Nevertheless, I consider this a very worthwhile study. Unlike some other studies, this study includes a sensitivity run (the E63H23-10CC run) that helps to assess a key uncertainty (partially re-iterating a point made in an earlier study). It also includes another useful sensitivity run (E63H23-LL) that helps to attribute differences between model versions to an individual change in a parameterization. In my opinion, after some revisions, this study clearly deserves to be published in GMD.

Specific comments and suggestions:

1. The fact that the CDNC threshold which leads to a lower ERFari+aci and a lower ECS is still applied should in my opinion definitely be mentioned in the abstract. As is shown in the manuscript, reducing this threshold results in a considerably larger ERFari+aci (first noted in Hoose et al., 2009) and interestingly also a larger ECS. It is not clear to me by how much an improved aerosol would reduce this larger ERFari+aci.

2. The weaker shortwave ERFari+aci in E63H23 is attributed to the new aerosol activation scheme and sea salt emission parameterization in E63H23 and a more realistic simulation of cloud water. Would it be possible to quantify individual contributions e.g. based on the E63H23-LL sensitivity study from Fig. S5 and perhaps also a run from Tegen et al. (2019)? Or would this require additional sensitivity runs?

3. It is concluded (p. 1, l. 27f) that "[t]he decrease in ECS in E63H23 (2.5 K) compared to E61H22 (2.8 K) is due to changes in the entrainment rate for shallow convection (affecting the cloud amount feedback) and a stronger cloud phase feedback". As far as I can see, especially the conclusion regarding a "stronger cloud phase feedback" (see also p. 24, l. 3) does not seem to be supported by sufficient evidence. Please either explain the existing evidence better, present additional evidence, or else please either preferably remove the statement or at least re-formulate the statement to reflect that this is not a finding but a speculation.

4. Cloud properties and cloud radiative effects are not only simulated by ECHAM-HAM but also by ECHAM. I wonder whether it would make sense to include ECHAM results in these comparisons. In my opinion, including ECHAM would in general help to better understand which biases in ECHAM-HAM are inherited and which biases are newly introduced by using e.g. different tunings and a different microphysics scheme. Including ECHAM results would also help to understand which differences between versions are due to changes in ECHAM and which differences between versions are due to changes in components that are specific to ECHAM-HAM. The discussion explaining the results frequently refers to changes in ECHAM, and some parts of it might be easier to follow if these changes were shown in tables 2 and 3 and especially also in the plots. On the other hand, the comparison of different ECHAM-HAM versions is useful without an additional focus on attributing the changes to changes in either standard ECHAM or in components that are specific to ECHAM-HAM, and including to many plots would also distract the reader from this comparison. Nevertheless, I think ECHAM plots would potentially be a nice-to-have. In case the authors decide against including ECHAM results, I would recommend to refer even more frequently to the literature documenting these results, especially when common biases are discussed. In some important cases (e.g. p. 20, l. 26), the references are already included.

Other specific comments and suggestions:

p. 1, l. 18ff "Biases that were identified in E63H23 (and in previous model versions) are a too low cloud amount in stratocumulus regions, deep convective clouds in the Atlantic and Pacific oceans form too close to the continents and there are indications that ICNCs are overestimated": I think that already here it would be good to clearly differentiate between biases that are inherited from ECHAM, biases that are specific to ECHAM-HAM in all HAM versions, and biases that change by the HAM modifications. Also, I think it would be good to clarify which biases were newly identified in this study and which biases are well-known and long-standing biases. Perhaps this can be achieved almost without lengthening the abstract. The resulting sentences could for

example read: "Common biases in ECHAM and in ECHAM-HAM are ... . ICNCs are overestimated in ....". If the authors decide against including ECHAM plots (discussed in comment #4 above), as far as ECHAM biases are documented elsewhere, it would be sufficient to point to the corresponding literature somewhere in the text.

p. 1, l. 19f, p. 17, l. 17f, p. 23, l. 12f: Based on Figs., 3, 6, and 8, I am not completely sure what is meant by "deep convective clouds in the Atlantic and Pacific oceans form too close to the continents". Is this something that one sees when putting ECHAM results next to ECHAM-HAM results? For example, there seems to be little deep convection over Indonesia in ECHAM and in ECHAM-HAM. Focusing on Indonesia (I think), Mauritsen et al., 2012 (https://doi.org/10.1029/2012MS000154) note that "[a]n interesting and challenging issue in MPI-ESM is the Tropical precipitation distribution over land versus ocean. The model prefers precipitating in the ocean, whereas observations indicate a stronger preference to precipitate on land."

p 1, l. 19f: in the Atlantic and Pacific oceans -> over the Atlantic and Pacific oceans

p. 2, l. 4: resulting in -> substantially contributing

p. 2, l. 5: realistic representation -> increasingly realistic representation (Almost certainly some of the dynamic responses to increased aerosol take place on scales which are too small to resolve by present-day global climate models. This remains a major concern as several studies point out that this potentially leads to an overestimate of ERFaci in coarse-resolution models.)

p. 4, l. 5: how does evaporation and sublimation affect aerosol number concentration?

p. 4, l. 28: I don't understand how CDNC from convective clouds is determined. Please try to explain this better.

p. 6, l. 9: please indicate that SALSA2.0 is not used here to avoid confusion.

p. 6, l. 18: please refer to my comment regarding p. 4, l. 28.

p. 7, l. 8: could you please be slightly more specific?

p. 8, l. 19: climate system -> atmosphere (if the entire climate system including the ocean were considered, 20 years would be insufficient. Using identical fixed SSTs strongly reduces the influence of internal variability.)

p. 11, l. 31: over -> cover

p. 13, l. 7f: The COSP CALIPSO simulator was used here, right? Excluding areas where the cloud products differ by more than five percentage points looks like a good idea to me. The excluded regions are regions in which one might expect problems.

p. 14, l. 8f: please refer to my comment regarding p. 4, l. 28.

p. 15, l. 32ff: In my opinion this entire discussion would be more interesting if standard ECHAM was included in the comparison.

p. 16, l. 27: where -> which

p. 16, l. 11f: the authors could mention that such a bias is also found in other models (https://doi.org/10.1029/2012GL053421).

p. 17, l. 6: RMS -> RMS error

p. 18, l. 22: does the statement "due to a smaller gamma_r" on p. 18, l. 22 indicate that the E63H23-10CC has been retuned? If yes, is this retuning expected to affect ECS? Based on https://doi.org/10.1175/JCLI-D-16-0151.1, I would have perhaps thought brighter clouds in the base state to play a different role for ECS. How about ER-Fari+aci? I wonder whether the result of Lohmann and Ferrachat (2010) holds also in E63H23.

p. 19, l. 26: why "...(Schmidt et al., 2006; Hansen et al., 2005) or Fan et al. (2004)"?

p. 21, lines 4 to 6.: is there a reference for this?

p. 22, l. 22: did ECS also decrease in standard ECHAM?

p. 23, l. 12: please refer to my comment regarding p. 1, l. 19f above.

p. 23, l. 21ff: please refer to my comment #2 above.

p. 24, l. 4: please refer to my comment #3 above.

Table 1: what exactly is gamma_r?

Table 3: from E61H22 to E63H23 there is a large compensation of changes in SW and LW ERFari+aci. This has also been noted elsewhere (https://doi.org/10.1002/2016GL071975). Are you really sure that here it is mainly due to the removal of a bug?

Fig. 9 is too small. It can't be enlarged because the resolution is too low. Please increase the size and the resolution.

Fig. 10: I do not understand the rationale behind excluding areas with little precipitation. While relative errors tend to be large in these areas, absolute errors in these areas tend to be small. Excluding these areas in the bias calculation could in principle hide model deficiencies. Observations of small values may contain important information due to a small absolute error, even where the relative error is large. Using standard deviations already ensures that large relative errors in regions with small values and small absolute errors will not have an overly large influence on the comparison.

Fig. 11: please increase the font size of the variable names (right plot title) and the color bar labels.

Fig. 14: Amt is amount, right? Please state this somewhere.

[Figure]

---

## Referee Comment (RC2) · Anonymous Referee #2 · 27 Apr 2019

General comments

The authors present an evaluation of clouds and precipitation simulated in three models ECHAM6.3-HAM2.3, ECHAM6.1-HAM2.2 and ECHAM5.5- HAM2.0 in comparison to global observational datasets. They discuss the performance of each model and the reasons for improvements.

The purpose of the paper is to provide a model documentation in a nutshell and to characterize the quality of clouds and precipitation simulated in the lates model E63H23 as well as in earlier versions. This is valuable information for all users and developers of these models and thus fits well to the scope of GMD. Overall the paper is clearly

structured and well written. Figures are of good quality.

My main recommendation is to add discussion to the obvious deviation between observed and modeled IWP near the equator and also in the NH storm tracks. Otherwise only minor modifications and corrections are necessary, so that a minor revision seems sufficient before publication.

Specific comments

Abstract, L14: low cloud amount → amount of low clouds (?)

P2 L1: "... Also, the spatial structure of multiple clouds shows a large variability on different scales as it depends not only on large scale motions of the air but also on convective and turbulent motions at different scales. ... " The important point is that the strong diabatic cooling/heating occurring with phase changes of water vapor causes a tight coupling between clouds and circulation, which is much less the case for other constituents.

P4 L12: orographic cirrus cloud → orographic cirrus clouds (2x)

P5-7 Section "2.1.5 Changes and improvements in E63H23" The preceding section lists already some process models and indicates cases where these are optional but not used in this study. In section 2.1.5 it is not pointed out that SALSA is available but – to my understanding – not used in this study. Is is also the case for other process models, for which improvements are listed here?

P7 L20: use the year → used the year

P7 L30: ... use a climatology for monthly values of sea surface temperature (SST) and sea ice cover (SIC) ... This setup excludes the influences of El Niño/La Niña on the variability of the atmospheric circulation. Thus the simulated "internal climate variability" is reduced compared to simulations which included El Niño/La Niña. This should be made clear to the reader. This reduction of variability is relevant for the later evaluation, see your comments on P8L18: "... to increase the signal of ERFari+aci

compared to variations in TOA net radiative flux due to internal variability of the climate system. . . . "

P8 L1: . . . the default configuration of these model versions . . . → . . . the default configuration of these ECHAM-HAM model versions . . . ECHAM6 on its own uses 47 levels (Stevens et al., 2013). P9 L17: . . . are described in Mauritsen et al. (2012) . . . The recent publication Mauritsen et al. (2019) provides more information for the tuning of ECHAM6.3: Mauritsen, T., Bader, J., Becker, T., Behrens, J., Bittner, M., Brokopf, R., et al. (2019). Developments in the MPI-M Earth System Model version 1.2 (MPI-ESM1.2) and its response to increasing CO2. Journal of Advances in Modeling Earth Systems, 11. https:// doi.org/10.1029/2018MS001400

P10 L26: . . . Table 3 . . . compared to observations (OBS) or multi-model mean values (MMM) when observations are not available . . . It would be helpful to mark each entry in the OBS/MMM column whether it is OBS or MMM.

P11 L32: cloud over → cloud cover

P12 L12: . . . The underestimation is particularly large in the tropics. . . . Here the authors should add explanations. What is the reason for this major deviation? Microphysics of deep convection? Cirrus cloud processes? As the authors have expertise in this field they should discuss this obvious modeling problem.

P13 L12: . . . Over land in the Northern Hemisphere the models overestimate cloud cover . . . This seems not correct for the dry continental regions: Sahara, Australia, . . ., even when accounting for less observational certainty in the Sahara region. Here the models underestimate the cloud cover.

P14 L12: . . . and in all model version shallow convection is triggered frequently . . . What is the role of the markedly increased shallow convection entrainment rate used in E63H23?

P14 L28: . . . The regional distribution of IWP of all three model versions agrees in

general quite well with the observations. . . . I do not agree with this general statement. The general disagreement in the equatorial region and also in the northern storm tracks is clear. Here, or earlier in the presentation of the zonal mean results, there is really the need to address this problem. I do not expect that a full explanations can be given – otherwise the modeling problem could be solved – but the authors should comment on this challenge and provide their insight in the possible reasons. This would make the discussion of the IWP much more interesting.

P15 L28: . . . the areas and magnitude of precipitation differs . . . → . . . the areas and magnitude of precipitation differ . . .

P17 L4: . . . the root-mean-square (RMS) error . . . → . . . the root-mean-square error (RMSE) . . . RMSE seems more useful for the following usage than RMS.

P20 L9: . . . The equilibrium climate sensitivity (ECS) is strongest in E55H20 (3.5 K), weaker in E61H22 (2.8 K) and weakest in E63H23 (2.5 K) (Fig. 13). . . . Here it would be valuable to have explained also the ECS values estimated for the base atmospheric models (ECHAM5, ECHAM6.1 and ECHAM6.3), as discussed in the literature. This would provide a better background for the discussion of the ECS estimates from the ECHAM-HAM models presented here. The recent Mauritsen et al (2019) paper also provides more information to the sensitivity of the ECS to certain model modifications.

---

## Author Response (AR1)

**Response to comments on *The global aerosol-climate model ECHAM6.3-HAM2.3 – Part 2: Cloud evaluation, aerosol radiative forcing and climate sensitivity*, Geosci. Model Dev. Discuss., gmd-2018-307**

David Neubauer, Sylvaine Ferrachat, Colombe Siegenthaler-Le Drian, Philip Stier, Daniel G. Partridge, Ina Tegen, Isabelle Bey, Tanja Stanelle, Harri Kokkola, and Ulrike Lohmann

We would like to thank the reviewers for the helpful comments and suggestions. They have helped to improve the content of the paper.
The original comments are in black. Responses are in blue. *Modifications to the text are in green and italics.*

**Anonymous Referee #1**

Neubauer et al. evaluate important aspects of a new version of a three-dimensional global aerosol-atmosphere model. It is well understood that cloud condensation nuclei concentrations vary significantly between strongly and less strongly polluted conditions and that radiative forcing by cloud-aerosol interactions depends non-linearly on cloud condensation nuclei concentration. At a time when box-model studies of cloud-aerosol interactions inform energy budget box-model studies, I find this study by Neubauer et al. a truly laudable effort. My only major concern with this study is that even in the latest model version, the minimum CDNC number is still artificially set to 40 per cubic centimetre. The authors discuss this point in some detail. They show that ERFari+aci and ECS depend on this threshold, but they do not mention this result in the abstract. Although one can certainly argue about this point, I find the reasons for applying this particular threshold unconvincing. Nevertheless, I consider this a very worthwhile study. Unlike some other studies, this study includes a sensitivity run (the E63H23-10CC run) that helps to assess a key uncertainty (partially re-iterating a point made in an earlier study). It also includes another useful sensitivity run (E63H23-LL) that helps to attribute differences between model versions to an individual change in a parameterization. In my opinion, after some revisions, this study clearly deserves to be published in GMD.

Specific comments and suggestions:

1. The fact that the CDNC threshold which leads to a lower ERFari+aci and a lower ECS is still applied should in my opinion definitely be mentioned in the abstract. As is shown in the manuscript, reducing this threshold results in a considerably larger ERFari+aci (first noted in Hoose et al., 2009) and interestingly also a larger ECS. It is not clear to me by how much an improved aerosol would reduce this larger ERFari+aci.

It is now mentioned in the abstract that the value do CDNCmin has an impact on ECS, as well as ERFari+aci:

*Experiments with minimum cloud droplet number concentrations (CDNCmin) of 40 cm-3 or 10 cm-3 show that a higher value of CDNCmin reduces ERFari+aci as well as ECS in E63H23.*

The experiment GFAS34 gives an indication by how much ERFari+aci would be reduced by an improved aerosol representation in ECHAM-HAM. E63H23-GFAS34 has a 17% lower ERFari+aci than E63H23. Since nitrate aerosol and secondary organic aerosol will occur in further regions, not only where wildfire emission occur, an even stronger decrease than 17% in ERFari+aci by an improved aerosol representation in ECHAM-HAM can be expected.

2. The weaker shortwave ERFari+aci in E63H23 is attributed to the new aerosol activation scheme and sea salt emission parameterization in E63H23 and a more realistic simulation of cloud water. Would it be possible to quantify individual contributions e.g. based on the E63H23-LL sensitivity study from Fig. S5 and perhaps also a run from Tegen et al. (2019)? Or would this require additional sensitivity runs?

We made another sensitivity experiment with E63H23 and the Guelle sea salt parameterization used in E61H22, named E63H23-GUELLE. Using the experiments E63H23, E63H23-LL, E63H23-GUELLE and E61H22 allows to estimate the individual contributions to the change in ERFari+aci from E61H22 to E63H23. But the different code/parameterization changes will not add linearly and there can be interaction between parameterizations, therefore the individual contributions are hard to disentangle. Over land the stronger ERFari+aci is caused about half by the new aerosol activation scheme and about half by other code changes.
Code changes (in particular the ICNC bugfix) cause a decrease by 0.65 Wm$^{-2}$ in LW ERFari+aci and the new aerosol activation scheme causes an increase by about 0.15 Wm$^{-2}$.
Over ocean the new aerosol activation scheme makes LW ERFari+aci stronger but has a weak impact on SW ERFari+aci. The new sea salt emission parameterization makes SW ERFari+aci stronger over ocean by about -0.1 Wm$^{-2}$ but the majority of the decrease in SW ERFari+aci over ocean by about 0.7 Wm$^{-2}$ in SW ERFari+aci is from other code changes (including e.g. the improved stratocumulus cloud cover in the base model ECHAM6.3 or tuning in ECHAM6.3).
We added the results of the E63H23-GUELLE sensitivity experiment to Table S1 and Fig. S5. We updated the text to better quantify individual contributions to changes in ERFari+aci between E61H22 and E63H23:

*A sensitivity simulation with the Lin and Leaitch (1997) activation scheme applied in E63H23 shows an ERFari+aci of 0.4 Wm-2 over land, explaining about half of the difference in ERFari+aci over land between E61H22 and E63H23.*

*The higher natural background aerosol concentrations due to the higher sea salt aerosol number concentrations in E63H23 explain also why ERFari+aci is less negative in E63H23 between 15°N and 45°N over oceans than in E61H22 (Fig. S5). From sensitivity simulations with the Lin and Leaitch (1997) activation scheme or Guelle et al. (2001) sea salt parameterization applied in E63H23 (Table S1) we conclude that the largest part of the change in SW ERFari+aci is actually from changes in the base model ECHAM6.3.*

3. It is concluded (p. 1, l. 27f) that "[t]he decrease in ECS in E63H23 (2.5 K) compared to E61H22 (2.8 K) is due to changes in the entrainment rate for shallow convection (affecting the cloud amount feedback) and a stronger cloud phase feedback". As far as I can see, especially the conclusion regarding a "stronger cloud phase feedback" (see also p. 24, l. 3) does not seem to be supported by sufficient evidence. Please either explain the existing evidence better, present additional evidence, or else please either preferably remove the statement or at least re-formulate the statement to reflect that this is not a finding but a speculation.

In Fig. 14 it can be seen that the largest difference in cloud feedback between E63H23 and E63H23-10CC is from a different cloud optical depth feedback. The change in non-low clouds optical depth feedback is likely due to a change in the cloud phase feedback. The new Fig. S7 shows the change in in-cloud CDNC from the 1xCO2 to the 2xCO2 climate. Cloud droplets exist at higher altitudes in the warmer climate. The increase in CDNC is stronger in E23H23 than in E63H23-10CC, leading to a larger change (increase) in cloud optical depth in the warmer climate in E63H23 compared to E63H23-10CC.

[Figure]

**Figure S7: Zonal mean change in in-cloud CDNC for E63H23 and E63H23-10CC for the change from the 1xCO$_2$ climate to the 2xCO$_2$ climate. Averages for the last 25 years of the 1xCO$_2$ and 2xCO$_2$ experiments were used to create the figure.**

We updated the text to refer to the new Fig. S7:
*The optical depth feedback of non-low clouds is less negative in the tropics and in mid-latitudes (not shown). This could be an indication of a weaker cloud phase feedback. As there are fewer but larger cloud droplets in E63H23-10CC than in E63H23 (Fig. S7), the cloud droplets have a shorter lifetime and this decreases differences between ice clouds and liquid water clouds.*

4. Cloud properties and cloud radiative effects are not only simulated by ECHAM-HAM but also by ECHAM. I wonder whether it would make sense to include ECHAM results in these comparisons. In my opinion, including ECHAM would in general help to better understand which biases in ECHAM-HAM are inherited and which biases are newly introduced by using e.g. different tunings and a different microphysics scheme. Including ECHAM results would also help to understand which differences between versions are due to changes in ECHAM and which differences between versions are due to changes in components that are specific to ECHAM-HAM. The discussion explaining the results frequently refers to changes in ECHAM, and some parts of it might be easier to follow if these changes were shown in tables 2 and 3 and especially also in the plots. On the other hand, the comparison of different ECHAM-HAM versions is useful without an additional focus on attributing the changes to changes in either standard ECHAM or in components that are specific to ECHAM-HAM, and including to many plots would also distract the reader from this comparison. Nevertheless, I think ECHAM plots would potentially be a nice-to-have. In case the authors decide against including ECHAM results, I would recommend to refer even more frequently to the literature documenting these results, especially when common biases are discussed. In some important cases (e.g. p. 20, l. 26), the references are already included.

The focus of the manuscript is on the comparison of different ECHAM-HAM versions and the inclusion of the latest version of the base model ECHAM6.3 in the Figures and Tables would indeed be a distraction for the reader as there are differences between ECHAM6.3 and E63H23 which would need to be discussed. We added more references discussing biases in cloud representation in the base model ECHAM (see answer to the next specific comment).
A comparison of the representation of clouds in ECHAM and ECHAM-HAM should be done in a future study.

Other specific comments and suggestions:

p. 1, l. 18ff "Biases that were identified in E63H23 (and in previous model versions) are a too low cloud amount in stratocumulus regions, deep convective clouds in the Atlantic and Pacific oceans form too close to the continents and there are indications that ICNCs are overestimated": I think that already here it would be good to clearly differentiate between biases that are inherited from ECHAM, biases that are specific to ECHAM-HAM in all HAM versions, and biases that change by the HAM modifications. Also, I think it would be good to clarify which biases were newly identified in this study and which biases are well-known and long-standing biases. Perhaps this can be achieved almost without lengthening the abstract. The resulting sentences could for example read: "Common biases in ECHAM and in ECHAM-HAM are ... . ICNCs are overestimated in ....". If the authors decide against including ECHAM plots (discussed in comment #4 above), as far as ECHAM biases are documented elsewhere, it would be sufficient to point to the corresponding literature somewhere in the text. p. 1, l. 19f, p. 17, l. 17f, p. 23, l. 12f: Based on Figs., 3, 6, and 8, I am not completely sure what is meant by "deep convective clouds in the Atlantic and Pacific oceans form too close to the continents". Is this something that one sees when putting ECHAM results next to ECHAM-HAM results? For example, there seems to be little deep convection over Indonesia in ECHAM and in ECHAM-HAM. Focusing on Indonesia (I think), Mauritsen et al., 2012 (https://doi.org/10.1029/2012MS000154) note that "[a]n interesting and challenging issue in MPI-ESM is the Tropical precipitation distribution over land versus ocean. The model prefers precipitating in the ocean, whereas observations indicate a stronger preference to precipitate on land."

We added references for the base model ECHAM at the suggested places, in particular:

Section 3.3:
*ECHAM underestimates tropical precipitation over land and overestimates tropical precipitation over ocean (Mauritsen et al., 2012; Stevens et al., 2013). This bias can be seen in Fig. 8 also for all ECHAM-HAM versions. Since ECHAM and ECHAM-HAM use the same parameterizations for convective clouds, this bias is very likely inherited from the base model ECHAM.*

Section 3.4:
*Deep convective clouds over the Atlantic and Pacific oceans form too close to the continents (see Figs. 3, 6 and 8) in E63H23 and ECHAM (Stevens et al., 2013). Stevens et al. (2013) speculate that the overestimation of tropical precipitation in the Atlantic close to South America may be related to the deficit of precipitation over South America in the tropics.*

In Figs. 3, 6 and 8 one can see in the observations that the maximum of cloud cover, IWP and precipitation in the ITCZ in the Pacific and the Atlantic is away from continents. Whereas in the different ECHAM-HAM versions (and also ECHAM). The maximum is of cloud cover, IWP and precipitation in the ITCZ in the Pacific and the Atlantic is close to continents.

Stevens et al. (2013) speculate for the Atlantic that the overestimation of tropical precipitation in the Atlantic close to South America may be related to the deficit of precipitation over South America in the tropics.

p 1, l. 19f: in the Atlantic and Pacific oceans -> over the Atlantic and Pacific oceans

Changed as suggested.

p. 2, l. 4: resulting in -> substantially contributing

Changed as suggested.

p. 2, l. 5: realistic representation -> increasingly realistic representation (Almost certainly some of the dynamic responses to increased aerosol take place on scales which are too small to resolve by present-day global climate models. This remains a major concern as several studies point out that this potentially leads to an overestimate of ERFaci in coarse-resolution models.)

We agree that this is a concern for global aerosol-climate models (e.g. Mülmenstädt and Feingold, 2018) and changed the phrase as suggested.

p. 4, l. 5: how does evaporation and sublimation affect aerosol number concentration?

A downward scavenging tracer flux is computed for each model column each model time step. In-cloud and below cloud scavenging are sources for the downward scavenging tracer flux, while evaporation and sublimation of precipitation are sinks for the downward scavenging tracer flux. When the sink term is larger than the source term of the downward scavenging tracer flux in a model level, aerosol mass and number concentrations will be transferred to the respective atmospheric tracers i.e. aerosol is released at this model level below clouds.
CDNC and ICNC used for wet scavenging are after cloud droplet evaporation, ice crystal sublimation and precipitation formation are computed. This is now mentioned in the description and an error in the description of nucleation scavenging has been corrected:
*The below cloud collection efficiencies as a function of aerosol and rain drop or snow crystal size are read from a look-up table. The in-cloud scavenging scheme takes nucleation scavenging and impaction scavenging of aerosol particles with cloud droplets and ice-crystals into account. For nucleation scavenging the number of scavenged aerosol particles is computed for liquid cloud droplets from the cloud droplet number concentration (CDNC) (after computation of cloud droplet evaporation and precipitation formation) and the fraction of activated aerosol particles computed by the activation scheme. For ice crystals the aerosol particles are scavenged progressively from the largest to the smallest modes until the number concentration of scavenged aerosol particles is equal to the ice crystal number concentration (ICNC) (after the computation of ice crystal sublimation and precipitation formation) of the grid box.*
*A downward scavenging tracer flux is computed for each model column each model time step. In-cloud and below cloud scavenging are sources for the downward scavenging tracer flux, while evaporation and sublimation of precipitation are sinks for the downward scavenging tracer flux. When the sink term is larger than the source term of the downward scavenging tracer flux in a model level, aerosol mass and number concentrations will be transferred to the respective atmospheric tracers i.e. aerosol is released from evaporating/sublimating precipitation at this model level back to the atmosphere.*

p. 4, l. 28: I don't understand how CDNC from convective clouds is determined. Please try to explain this better.

A description of how CDNC from convective clouds are computed has been moved from section 3.3 to section 2.1.3 and extended:

*To obtain CDNC for the detrained condensate, several simplifications are applied. It is assumed that cloud droplets of convective clouds will form at cloud base. The number of activated cloud condensation nuclei (CCN) at the convective cloud base is computed using the vertical velocity from large scale and turbulent fluxes as described in section 2.1.4. It is further assumed that CDNC will be constant throughout the vertical extension of the convective clouds. At the level of detrainment these CDNC from the convective clouds will either evaporate or be added to stratiform clouds if these exist at the level of detrainment. In the latter case a weighted average of the stratiform CDNC and detrained CDNC is computed by weighting stratiform CDNC with the stratiform liquid water content and detrained CDNC with detrained liquid water content. CDNC of the stratiform cloud is not allowed to decrease by this procedure, since cloud droplets will not evaporate in a supersaturated environment.*

p. 6, l. 9: please indicate that SALSA2.0 is not used here to avoid confusion.

Done.

p. 6, l. 18: please refer to my comment regarding p. 4, l. 28.

Section 2.1.5 list changes and improvements in E63H23 but is not meant to provide the details of the changes which are given in other sections (section 2.1.3 for the computation of CDNC from convective clouds). Nevertheless this point was reformulated to avoid confusion:

*Changed treatment of detrained cloud water mass and number concentrations from convective clouds: CDNC from detrained cloud water added (weighted average) to CDNC of a stratiform cloud cannot decrease CDNC of the stratiform cloud; split between liquid water and ice of detrained condensate is made consistent between mass and number concentrations*

p. 7, l. 8: could you please be slightly more specific?

We list now the most important updates:

*Update of default settings, run templates and run organization: the vertical resolution is per default 47 vertical model layers; the reference year and reference period for present day simulations are 2008 and 2003-2012 respectively*

p. 8, l. 19: climate system -> atmosphere (if the entire climate system including the ocean were considered, 20 years would be insufficient. Using identical fixed SSTs strongly reduces the influence of internal variability.)

That's a good point. We changed this to:

*The simulation time was 20 years to increase the signal of ERFari+aci compared to variations in TOA net radiative flux due to internal variability of the atmosphere. The use of an identical climatology for SST/SIC in all simulations reduces the internal variability compared to simulations with a global climate model (GCM) coupled to a full ocean model.*

p. 11, l. 31: over -> cover

Done.

p. 13, l. 7f: The COSP CALIPSO simulator was used here, right? Excluding areas where the cloud products differ by more than five percentage points looks like a good idea to me. The excluded regions are regions in which one might expect problems.

The COSP simulator was not implemented in E55H20 (mentioned in section 3.3). Therefore Fig. 3 shows the direct model output without using the COSP CALIPSO simulator. This is now mentioned in the figure caption of Fig. 3:

*… model data is from the PD simulations (direct model output is used without a simulator).*

The new Fig. S8 shows the cloud cover from the COSP CALIPSO simulator implemented in E61H22 and E63H23 and a comparison to the direct model output of the cloud cover. Except at high latitudes (where observations are uncertain) and a few areas, there are no large differences between the direct model output and the COSP CALIPSO simulator output:

*Since the COSP CALIPSO simulator is not implemented in E55H20, the direct model output is shown for all model versions (see Fig. S8 for COSP CALIPSO simulator output of cloud cover for E61H22 and E63H23).*

[Figure]

**Figure S8: Comparison of annual mean cloud cover of E55H20, E61H22 and E63H23 to CALIPSO GOCCP observations. Areas where the cloud cover of CALIPSO GOCCP, MODIS collection 6.1 and AVHRR-PM differ by more than five percent points are hatched. CALIPSO GOCCP data is for 2006-2010, model data is from the PD simulations. For E61H22 and E63H23 the direct model output, the output of the COSP CALIPSO simulator implemented in those model versions and the difference between direct model output and COSP CALIPSO simulator output is displayed.**

p. 14, l. 8f: please refer to my comment regarding p. 4, l. 28.

The description how detrained CDNC are computed and added to an existing stratiform cloud was moved to section 2.1.3 (see our answer to the comment regarding p. 4, l. 28):
*The weighted average of stratiform CDNC and detrained CDNC (see section 2.1.3) may overestimate the CDNC concentration of shallow cumulus clouds.*

p. 15, l. 32ff: In my opinion this entire discussion would be more interesting if standard ECHAM was included in the comparison.

See our answer to your comment #4.

p. 16, l. 27: where -> which

Done.

p. 16, l. 11f: the authors could mention that such a bias is also found in other models (https://doi.org/10.1029/2012GL053421).

Thanks, this has been added.

p. 17, l. 6: RMS -> RMS error

Done.

p. 18, l. 22: does the statement "due to a smaller gamma_r" on p. 18, l. 22 indicate that the E63H23-10CC has been retuned? If yes, is this retuning expected to affect ECS? Based on https://doi.org/10.1175/JCLI-D-16-0151.1, I would have perhaps thought brighter clouds in the base state to play a different role for ECS. How about ERFari+aci? I wonder whether the result of Lohmann and Ferrachat (2010) holds also in E63H23.

E63H23-10CC has been retuned. The value of gamma_r has been added to the text:
*(due to retuning with a smaller $\gamma_r=2.8$; not shown)*
The study by Klocke et al. (2011), identified the entrainment rate for shallow convection and the cloud mass flux above the level of neutral buoyancy as tuning factors in ECHAM that affect ECS. They did not test the conversion rate from cloud water to rain in stratiform clouds but the conversion rate from cloud water to rain for convective clouds and found no significant impact on ECS. The results in Figure 6 of Lohmann and Neubauer (2018) indicate no large impact of the tuning parameter gamma_r on cloud feedback parameters and therefore ECS.
The autoconversion parameterization in E55H20 used by Lohmann and Ferrachat (2010) is the same as in E63H23, so the meaning of gamma_r and the response to changes in gamma_r should be similar. Investigating the effect of model tuning on ECS and ERFari+aci would be interesting but is beyond the focus of this study.

p. 19, l. 26: why "...(Schmidt et al., 2006; Hansen et al., 2005) or Fan et al. (2004)"?

We rephrased this part to make it clear that different models were used in these studies:
*... have been found by Fan et al. (2004) (using the Geophysical Fluid Dynamics Laboratory (GFDL) global chemical transport model; Mahlman and Moxim, 1978) and Bauer and Koch (2005) and Bauer et al. (2007) (using the Goddard Institute for Space Studies (GISS) climate model, modelE; Schmidt et al., 2006; Hansen et al., 2005).*

p. 21, lines 4 to 6.: is there a reference for this?

Thanks, unfortunately this was overlooked. These changes are described in Mauritsen et al. (2019), the reference was added.

p. 22, l. 22: did ECS also decrease in standard ECHAM?

Yes, we added ECS for the three corresponding ECHAM base model versions:
*The equilibrium climate sensitivity (ECS) is strongest in E55H20 (3.5 K), weaker in E61H22 (2.8 K) and weakest in E63H23 (2.5 K) (Fig. 13). The corresponding ECS values in the base model versions are: ECHAM5: 3.4 K (Randall et al., 2007; their Table 8.2), ECHAM6.1: 2.8 K (Block and Mauritsen, 2013; Meraner et al., 2013) and ECHAM6.3: 2.8 K (Mauritsen et al., 2019), i.e. changes in ECS between the ECHAM-HAM model versions are driven substantially by changes in the ECHAM base model versions. Note that the ECS value for ECHAM6.3 is from a simulation with abruptly quadrupled CO2 concentrations, in contrast to the CO2 doubling used in the computation of ECS in this study and that ECHAM has a strong state dependency for ECS (see discussion in Mauritsen et al., 2019 and references therein).*

We changed the sentence in p. 22, l: 21-22 to:
*The changes and improvements in E63H23 (including changes in the base model version ECHAM6.3) therefore have not only improved the representation of clouds in E63H23 compared to previous model versions, they also have an impact on ERFari+aci and ECS, decreasing both.*

p. 23, l. 12: please refer to my comment regarding p. 1, l. 19f above.

Changed as suggested.

p. 23, l. 21ff: please refer to my comment #2 above.

This was changed to better reflect the contribution of individual changes:
*The largest differences between E61H22 and E63H23 in terms of SW ERFari+aci are due to:*
- *mainly the more realistic simulation of cloud water,*
- *but also the new activation scheme in E63H23 and*
- *the new sea salt emission parameterization*

*which lead to a weaker SW ERFari+aci in E63H23. In terms of LW ERFari+aci the difference is mainly due to:*
- *the removal of an inconsistency in ICNC in cirrus clouds leading to a weaker LW ERFari+aci in E63H23.*

p. 24, l. 4: please refer to my comment #3 above.

See our answer to your comment #3.

Table 1: what exactly is gamma_r?

gamma_r is a unitless scaling factor for the stratiform rain formation rate by autoconversion. The header of Table 2 has been updated to reflect this:
*… scaling factor for stratiform rain formation rate by autoconversion ($\gamma_r$), scaling factor for stratiform snow formation rate by aggregation ($\gamma_s$),…*

Table 3: from E61H22 to E63H23 there is a large compensation of changes in SW and LW ERFari+aci. This has also been noted elsewhere (https://doi.org/10.1002/2016GL071975). Are you really sure that here it is mainly due to the removal of a bug?

Test simulations before and after the ICNC bugfix revealed a large change in LW ERFari+aci (not shown). The large change in LW ERFari+aci can be attributed mainly to this bugfix and changes in SW ERFari+aci to other model changes (see our answer to your comment #2).

Fig. 9 is too small. It can't be enlarged because the resolution is too low. Please increase the size and the resolution.

Done.

Fig. 10: I do not understand the rationale behind excluding areas with little precipitation. While relative errors tend to be large in these areas, absolute errors in these areas tend to be small. Excluding these areas in the bias calculation could in principle hide model deficiencies. Observations of small values may contain important information due to a small absolute error, even where the relative error is large. Using standard deviations already ensures that large relative errors in regions with small values and small absolute errors will not have an overly large influence on the comparison.

The areas hatched in Fig. 8 are indeed areas where the relative error is large but the absolute error is small. Therefore we recomputed the correlation and normalized standard deviation including all areas for precipitation. At the same time we corrected a mistake in the computation of the normalized standard deviation. The discussion of Fig. 10 has been adapted:
*The standardized deviations of LWP-LP had to be scaled by a factor of 1/4 so they could fit on the scale. For all variables the root-mean-square error (RMSE) (solid circles in the diagram in Fig. 10) is smaller or equal in E63H23 compared to in E61H22 and E55H20 (note the scaling for LWP-LP).*

[Figure]

**Figure 10: Taylor diagram for comparison of SW and LW CRE, cloud cover, LWP-low precipitation, Cloud top CDNC, IWP and precipitation of E55H20, E61H22 and E63H23 to observations as REF. The standardized deviations of LWP-low precipitation are scaled by a factor of 1/4 to fit on the diagram. Only areas that are not hatched in Figs. 3 - 6 were used to create the Taylor diagram. Observations are the same as in Figs. 2 - 6 and 8. The correlation coefficient is shown as an angle and quantifies the similarity in pattern between modelled and observed annual mean fields. The standard deviation of the modelled fields (normalized by the standard deviation of the observed fields) is shown as the radial distance from the origin. The RMSE is shown as solid black circles and is the distance from the point marked by REF (the closer a model is to REF the better its skill to reproduce the observations). For E63H23 and the observations for precipitation and LWP-low precipitation an average over the time period 2003 to 2012 was used. For Cloud top CDNC the time period 2003 to 2015, for IWP the time periods in Li et al. (2012), for SW CRE and LW CRE the time period July 2005 to June 2015, for cloud cover the time period June 2006 to December 2010 and for E55H20 and E63H23 the time period 2000 to 2009 were used. Tests with E63H23 showed negligible impact of the different time periods for the data in the Taylor diagram.**

Fig. 11: please increase the font size of the variable names (right plot title) and the color bar labels.

Done.

Fig. 14: Amt is amount, right? Please state this somewhere.

Done.

**Anonymous Referee #2**

General comments
The authors present an evaluation of clouds and precipitation simulated in three models ECHAM6.3-HAM2.3, ECHAM6.1-HAM2.2 and ECHAM5.5-HAM2.0 in comparison to global observational datasets. They discuss the performance of each model and the reasons for improvements.
The purpose of the paper is to provide a model documentation in a nutshell and to characterize the quality of clouds and precipitation simulated in the lates model E63H23 as well as in earlier versions. This is valuable information for all users and developers of these models and thus fits well to the scope of GMD. Overall the paper is clearly structured and well written. Figures are of good quality. My main recommendation is to add discussion to the obvious deviation between observed and modeled IWP near the equator and also in the NH storm tracks. Otherwise only minor modifications and corrections are necessary, so that a minor revision seems sufficient before publication.

We added a discussion of the underestimation of cloud ice in the tropics and refer to the discussion of the general low bias of IWP in ECHAM-HAM at the end of section 3.3. See our answers to the specific comments below.

Specific comments
Abstract, L14: low cloud amount → amount of low clouds (?)

Changed to amount of low clouds.

P2 L1: "...Also, the spatial structure of multiple clouds shows a large variability on different scales as it depends not only on large scale motions of the air but also on convective and turbulent motions at different scales. ..." The important point is that the strong diabatic cooling/heating occurring with phase changes of water vapor causes a tight coupling between clouds and circulation, which is much less the case for other constituents.

This is a very good point. We added this to the introduction:
*Also, the spatial structure of multiple clouds shows a large variability on different scales as it depends not only on large scale motions of the air but also on convective and turbulent motions at different scales. These convective and turbulent motions in turn are driven to a large part by diabatic heating (and cooling) and radiative cooling (and heating) involving cloud and precipitation hydrometeors, leading to a tight coupling between clouds and circulation.*

P4 L12: orographic cirrus cloud → orographic cirrus clouds (2x)

Done.

P5-7 Section "2.1.5 Changes and improvements in E63H23" The preceding section lists already some process models and indicates cases where these are optional but not used in this study. In section 2.1.5 it is not pointed out that SALSA is available but – to my understanding – not used in this study. Is is also the case for other process models, for which improvements are listed here?

It is now explicitly mentioned in section 2.1.5 that SALSA is not used in this study. All other changes and improvements in E63H23 listed in section 2.1.5 are used for the simulations with E63H23 in this study.

P7 L20: use the year → used the year

Done.

P7 L30: ... use a climatology for monthly values of sea surface temperature (SST) and sea ice cover (SIC) ... This setup excludes the influences of El Niño/La Niña on the variability of the atmospheric circulation. Thus the simulated "internal climate variability" is reduced compared to simulations which included El Niño/La Niña. This should be made clear to the reader. This reduction of variability is relevant for the later evaluation, see your comments on P8L18: " ... to increase the signal of ERFari+aci compared to variations in TOA net radiative flux due to internal variability of the climate system. ..."

That's a good point. We changed this to:
*The simulation time was 20 years to increase the signal of ERFari+aci compared to variations in TOA net radiative flux due to internal variability of the atmosphere. The use of an identical climatology for SST/SIC in all simulations reduces the internal variability compared to simulations with a global climate model (GCM) coupled to a full ocean model.*

P8 L1: ... the default configuration of these model versions ... → ... the default configuration of these ECHAM-HAM model versions ... ECHAM6 on its own uses 47 levels (Stevens et al., 2013).

Changed as suggested.

P9 L17: ... are described in Mauritsen et al. (2012) ... The recent publication Mauritsen et al. (2019) provides more information for the tuning of ECHAM6.3: Mauritsen, T., Bader, J., Becker, T., Behrens, J., Bittner, M., Brokopf, R., et al. (2019). Developments in the MPI-M Earth System Model version 1.2 (MPI-ESM1.2) and its response to increasing CO2. Journal of Advances in Modeling Earth Systems, 11. https:// doi.org/10.1029/2018MS001400

Thanks, this was changed to:
*The tuning strategy and parameters for ECHAM6 as well as the impact of these parameters on the model climate are described in Mauritsen et al. (2012, 2019).*

P10 L26: ... Table 3 ... compared to observations (OBS) or multi-model mean values (MMM) when observations are not available ... It would be helpful to mark each entry in the OBS/MMM column whether it is OBS or MMM.

Done.

P11 L32: cloud over → cloud cover

Done.

P12 L12: ... The underestimation is particularly large in the tropics. ... Here the authors should add explanations. What is the reason for this major deviation? Microphysics of deep

convection? Cirrus cloud processes? As the authors have expertise in this field they should discuss this obvious modeling problem.

We added a discussion of the underestimation of cloud ice in the tropics:
*The underestimation is particularly large in the tropics, which is likely connected to the parameterization of convection in ECHAM(-HAM). ECHAM has a low precipitation bias over land in the tropics (Mauritsen et al., 2012; Stevens et al., 2013). Gasparini et al. (2018) found indications that the level of detrainment from deep convection is too low in altitude in ECHAM-HAM. They lowered the tuning parameter for deep convective entrainment $\epsilon_d$ to 0.00006, whereas all three ECHAM-HAM versions used here have to use a larger value for this parameter (Table 2) as they use a cirrus scheme in which cirrus clouds can only nucleate homogeneously, which may lead to an overestimation of ICNC and underestimation of IWP by tuning of radiative fluxes (see section 3.3).*

P13 L12: ... Over land in the Northern Hemisphere the models overestimate cloud cover ... This seems not correct for the dry continental regions: Sahara, Australia, ... , even when accounting for less observational certainty in the Sahara region.   Here the models underestimate the cloud cover.

This was indeed imprecise and was changed to:
*Over land in the Northern Hemisphere polwards of about 45°N the models overestimate cloud cover ...*

P14 L12: ... and in all model version shallow convection is triggered frequently ... What is the role of the markedly increased shallow convection entrainment rate used in E63H23?

The change in shallow convection entrainment rate will shift the level of detrainment to lower altitudes. Since CDNC of detrained condensate are determined from convective cloud base, the CDNC of detrained condensate do not depend on the altitude of the level of detrainment.

P14 L28: ... The regional distribution of IWP of all three model versions agrees in general quite well with the observations. ... I do not agree with this general statement. The general disagreement in the equatorial region and also in the northern storm tracks is clear.  Here, or earlier in the presentation of the zonal mean results, there is really the need to address this problem. I do not expect that a full explanations can be given – otherwise the modeling problem could be solved – but the authors should comment on this challenge and provide their insight in the possible reasons.  This would make the discussion of the IWP much more interesting.

This statement was reformulated to clarify that the magnitude of IWP does not agree with observations but IWP occurs in the same areas as in observations. For the discussion of the low bias in the tropics we refer now to the discussion in section 3.2. A reason for the low bias of IWP is given at the end of section 3.3.
*The regional distribution of occurrence of IWP of all three model versions agrees in general quite well with the observations, although it is biased low in all ECHAM-HAM model versions. This could already be seen in the respective global mean and zonal mean values (see sections 3.1 and 3.2). Similar to what was found in the analysis of zonal mean IWP the underestimation is largest in the tropics (see section 3.2).*
We refer to this now also in section 3.4:
*Previous studies (Gasparini et al., 2018; Lohmann and Neubauer, 2018) showed that this overestimation of ICNC is (at least partly) due to missing processes in the formation of cirrus*

*clouds (heterogeneous freezing of ice nucleating particles and/or water vapor deposition on pre-existing ice crystals). These studies also showed that including these processes can reduce the underestimation of IWP in ECHAM-HAM.*

P15 L28: ... the areas and magnitude of precipitation differs ... → ... the areas and magnitude of precipitation differ ...

Thanks.

P17 L4: ... the root-mean-square (RMS) error ... → ... the root-mean-square error (RMSE) ... RMSE seems more useful for the following usage than RMS.

Changed as suggested.

P20 L9: ... The equilibrium climate sensitivity (ECS) is strongest in E55H20 (3.5 K), weaker in E61H22 (2.8 K) and weakest in E63H23 (2.5 K) (Fig. 13). ... Here it would be valuable to have explained also the ECS values estimated for the base atmospheric models (ECHAM5, ECHAM6.1 and ECHAM6.3), as discussed in the literature. This would provide a better background for the discussion of the ECS estimates from the ECHAM-HAM models presented here. The recent Mauritsen et al (2019) paper also provides more information to the sensitivity of the ECS to certain model modifications.

We added ECS for the three corresponding ECHAM base model versions:
*The equilibrium climate sensitivity (ECS) is strongest in E55H20 (3.5 K), weaker in E61H22 (2.8 K) and weakest in E63H23 (2.5 K) (Fig. 13). The corresponding ECS values in the base model versions are: ECHAM5: 3.4 K (Randall et al., 2007; their Table 8.2), ECHAM6.1: 2.8 K (Block and Mauritsen, 2013; Meraner et al., 2013) and ECHAM6.3: 2.8 K (Mauritsen et al., 2019), i.e. changes in ECS between the ECHAM-HAM model versions are driven substantially by changes in the ECHAM base model versions. Note that the ECS value for ECHAM6.3 is from a simulation with abruptly quadrupled $CO_2$ concentrations, in contrast to the $CO_2$ doubling used in the computation of ECS in this study and that ECHAM has a strong state dependency for ECS (see discussion in Mauritsen et al., 2019 and references therein).*
To our knowledge no ECS value for doubling of $CO_2$ using a mixed-layer ocean model of ECHAM6.3 have been published, therefore we refer to the state dependence of ECS in ECHAM, discussed in Mauritsen et al. (2019). Published ECS values of ECHAM5, ECHAM6.1, ECHAM6.3:
5.4 K (Li et al, 2012; ECHAM5/MPIOM 4xCO2; T31L19GR30; simulation thousands of years long)
5.6 K (Li et al, 2012; ECHAM5, 4xCO2 MLO; T31L19)
3.4 K (Randall et al., 2007; their Table 8.2; ECHAM5, 2xCO2 MLO; T63L31)

2.8 K (Block and Mauritsen, 2013; ECHAM6.1, 2xCO2 MLO)
2.8 K (Meraner et al., 2013; ECHAM6.1, 2xCO2 MLO)
3.4 K (Block and Mauritsen, 2013; ECHAM6.1, 4xCO2 MLO)
3.4/3.9 K (Meraner et al., 2013; ECHAM6.1, 4xCO2 MLO; first value from deltaTs change; second value from F/lambda)
3.7 K (Block and Mauritsen, 2013; MPI-ESM-LR (ECHAM6.1) 4xCO2; 150 years after CO2 quadrupling)
3.6 K (Flato et al., 2013; their Table 9.5; MPI-ESM-LR (ECHAM6.1) 4xCO2; 150 years after CO2 quadrupling)

3.7 K (Giorgetta et al. (2013); MPI-ESM-LR (ECHAM6.1) 4xCO2; 150 years after CO2 quadrupling)

2.8 K (Mauritsen et al, 2019; MPI-ESM1.2-LR 4xCO2; 150 years after CO2 quadrupling)
3.6 K (Mauritsen et al, 2019; MPI-ESM1.2-LR 4xCO2; 100-1000 years after CO2 quadrupling)
2.8 K (Mauritsen et al, 2019; MPI-ESM1.2-LR 2xCO2; 100-1000 years after CO2 quadrupling)

We added the reference to Mauritsen et al. (2019) for the change in the entrainment rate for shallow convection and its impact on ECS:
*This seems to be related to the stronger entrainment rate for shallow convection in E63H23 (Mauritsen et al., 2012, 2019). Mauritsen et al. (2019) describe the increase of entrainment rate for shallow convection in ECHAM6.3 as a measure to reduce ECS in ECHAM6.3.*

[revised manuscript text omitted]